# Yap1 safeguards mouse embryonic stem cells from excessive apoptosis during differentiation

Lucy LeBlanc[1,2], Bum-Kyu Lee[1,2], Andy C Yu[1], Mijeong Kim[1,2], Aparna V Kambhampati[1], Shannon M Dupont[1], Davide Seruggia[3,6,5], Byoung U Ryu[1], Stuart H Orkin[3,4,5,6], Jonghwan Kim[1,2]*

[1]Department of Molecular Biosciences, The University of Texas at Austin, Austin, United States; [2]Institute for Cellular and Molecular Biology, Center for Systems and Synthetic Biology, The University of Texas at Austin, Austin, United States; [3]Division of Hematology/Oncology, Boston Children's Hospital, Boston, United States; [4]Howard Hughes Medical Institute, Boston, United States; [5]Department of Pediatric Oncology, Dana-Farber Cancer Institute (DFCI), Boston, United States; [6]Harvard Stem Cell Institute, Harvard Medical School, Boston, United States

**Abstract** Approximately, 30% of embryonic stem cells (ESCs) die after exiting self-renewal, but regulators of this process are not well known. Yap1 is a Hippo pathway transcriptional effector that plays numerous roles in development and cancer. However, its functions in ESC differentiation remain poorly characterized. We first reveal that ESCs lacking Yap1 experience massive cell death upon the exit from self-renewal. We subsequently show that Yap1 contextually protects differentiating, but not self-renewing, ESC from hyperactivation of the apoptotic cascade. Mechanistically, Yap1 strongly activates anti-apoptotic genes via *cis*-regulatory elements while mildly suppressing pro-apoptotic genes, which moderates the level of mitochondrial priming that occurs during differentiation. Individually modulating the expression of single apoptosis-related genes targeted by Yap1 is sufficient to augment or hinder survival during differentiation. Our demonstration of the context-dependent pro-survival functions of Yap1 during ESC differentiation contributes to our understanding of the balance between survival and death during cell fate changes.
DOI: https://doi.org/10.7554/eLife.40167.001

*For correspondence:
jonghwankim@mail.utexas.edu

Competing interests: The authors declare that no competing interests exist.

## Introduction

Yap1 regulates genes involved in many cellular functions, including proliferation, organ size control, and tumorigenesis (*Ehmer and Sage, 2016*; *Hansen et al., 2015*; *Huang et al., 2005*). When Hippo signaling is active, kinases Lats1/2 phosphorylate Yap1, leading to cytoplasmic sequestration (*Hao et al., 2008*). When Hippo signaling is inactive, Yap1 translocates to the nucleus to co-activate or co-repress numerous target genes with interacting partner proteins such as Tead factors (*Kim et al., 2015*; *Stein et al., 2015*).

Previous research indicated that, in mouse embryonic stem cells (ESCs), nuclear translocation of Yap1 occurs shortly after withdrawal of leukemia inhibitory factor (LIF), a cytokine that maintains self-renewal, and that depletion of Yap1 inhibits differentiation, whereas overexpression (OE) of Yap1 stimulates differentiation (*Chung et al., 2016*). Deletion of Yap1 leads to embryonic lethality by E10.5 although the downstream mechanism remains poorly characterized (*Morin-Kensicki et al., 2006*). Additionally, whether Yap1 has any other roles during ESC differentiation and early development remains unclear.

Apoptosis influences numerous biological processes, including development, differentiation, and infection (*Fuchs and Steller, 2011*; *Meier et al., 2000*). A previous study has reported that withdrawal of LIF causes the death of 30% or more of ESCs (*Bashamboo et al., 2006*; *Duval et al., 2000*), and around 30% of human ESCs are also annexin V positive when they exit from self-renewal (*Dravid et al., 2005*). A proposed function of apoptosis during ESC differentiation is to cull cells that fail to exit self-renewal, thus promoting efficient differentiation (*Wang et al., 2015*). This process is not limited to ESCs, as defective cells are executed during human neural progenitor differentiation as well (*Jaeger et al., 2015*), and apoptosis eliminates self-reactive and non-reactive lymphocytes during T and B cell differentiation (*Francelin, 2011*; *Nemazee, 2017*; *Opferman, 2008*).

This process must be finely tuned to ensure efficient changes in cell identity without excessive loss of cell viability. However, mechanisms that regulate the balance between survival and death during ESC differentiation remain insufficiently characterized. Here, we find that Yap1 attenuates mitochondrial apoptosis during ESC differentiation, primarily by upregulating anti-apoptotic factors, such as Bcl-2, Bcl-xL (*Bcl2l1*), and Mcl-1, through direct transcriptional regulation. Mouse ESCs lacking Yap1 have no defect in survival in self-renewing conditions. However, just after the exit from self-renewal, we find that Yap1 knockout (KO) cells develop a high degree of mitochondrial priming that precedes elevated rates of apoptosis. OE of anti-apoptotic factors or repression of pro-apoptotic factors in Yap1 KO cells rescues this enhanced rate of cell death during differentiation. This collectively suggests that Yap1 is critical for ESC survival in a context-dependent manner, advancing our understanding of regulation of cell death during changes in cell identity.

## Results

### Genetic ablation of *Yap1* intensifies caspase-dependent cell death during ESC differentiation

To determine context-specific roles of Yap1, we attempted to differentiate J1 ESCs in which *Yap1* had been deleted via CRISPR/Cas9 in KO clones established in our previous publication (*Figure 1—figure supplement 1A*). While ~30% cell death was observed from wild-type (WT) cells as previously reported (*Bashamboo et al., 2006*), cell death was dramatically higher (up to >70%) in Yap1 KO cells 72 hr after LIF withdrawal (*Figure 1A* and *Figure 1—figure supplement 1B*). In both cases, cell death was substantially reduced after supplementation with Z-VAD-FMK (zVAD), a pan-caspase inhibitor, but not with necrostatin-1, which blocks necroptosis. Undifferentiated cells had extremely low rates of cell death regardless of genotype (*Figure 1A*). An additional J1 Yap1 KO clone as well as Yap1 KO clones established in the CJ7 and E14 ESC lines (*Figure 1—figure supplement 1C*) also experienced drastically heightened cell death during differentiation, but not self-renewal (*Figure 1B*). Furthermore, depletion of *Yap1* using shRNA-mediated knockdown (KD) dose-dependently increased cell death during differentiation (*Figure 1—figure supplement 1D and E*). Finally, measuring cell death in stable Yap1 OE cell lines (*Figure 1*-figure supplement F) reduced cell death to a mere ~10% during differentiation (*Figure 1C*). Thus, Yap1 is key for survival during ESC differentiation, and ablation of *Yap1* specifically exacerbates apoptosis.

### Loss of Yap1 leads to caspase hyperactivation during differentiation

During apoptosis, initiator caspases 8 (Casp8) and 9 (Casp9) are activated first, either by death receptors or mitochondrial outer membrane permeabilization, respectively (*Bao and Shi, 2007*). They then cleave executioner caspases such as caspase-3 (Casp3), which then cleave hundreds of downstream targets in the cell that result in its death, including Parp1 (*Fischer et al., 2003*). Treatment of ESCs with NucView 488 enabled live visualization of active Casp3. In undifferentiated ESCs, Casp3 activation was rare in both WT and KO cells, but the proportion of cells with active Casp3 increased visibly after LIF withdrawal as a function of time (*Figure 1D*). Notably, a far greater proportion of Yap1 KO cells than WT cells possessed active Casp3 by 60 hr. Then, we performed flow cytometry to quantify active Casp3 as well as externalized phosphatidylserine. Both the relative proportion of Casp3 positive cells and the fluorescent intensity of the Casp3 substrate fluorescent probe were higher in Yap1 KO differentiating ESCs (dESCs), and this was correlated with an increased proportion of annexin V positive cells (*Figure 1E and F*). Immunoblot analysis confirmed faster and

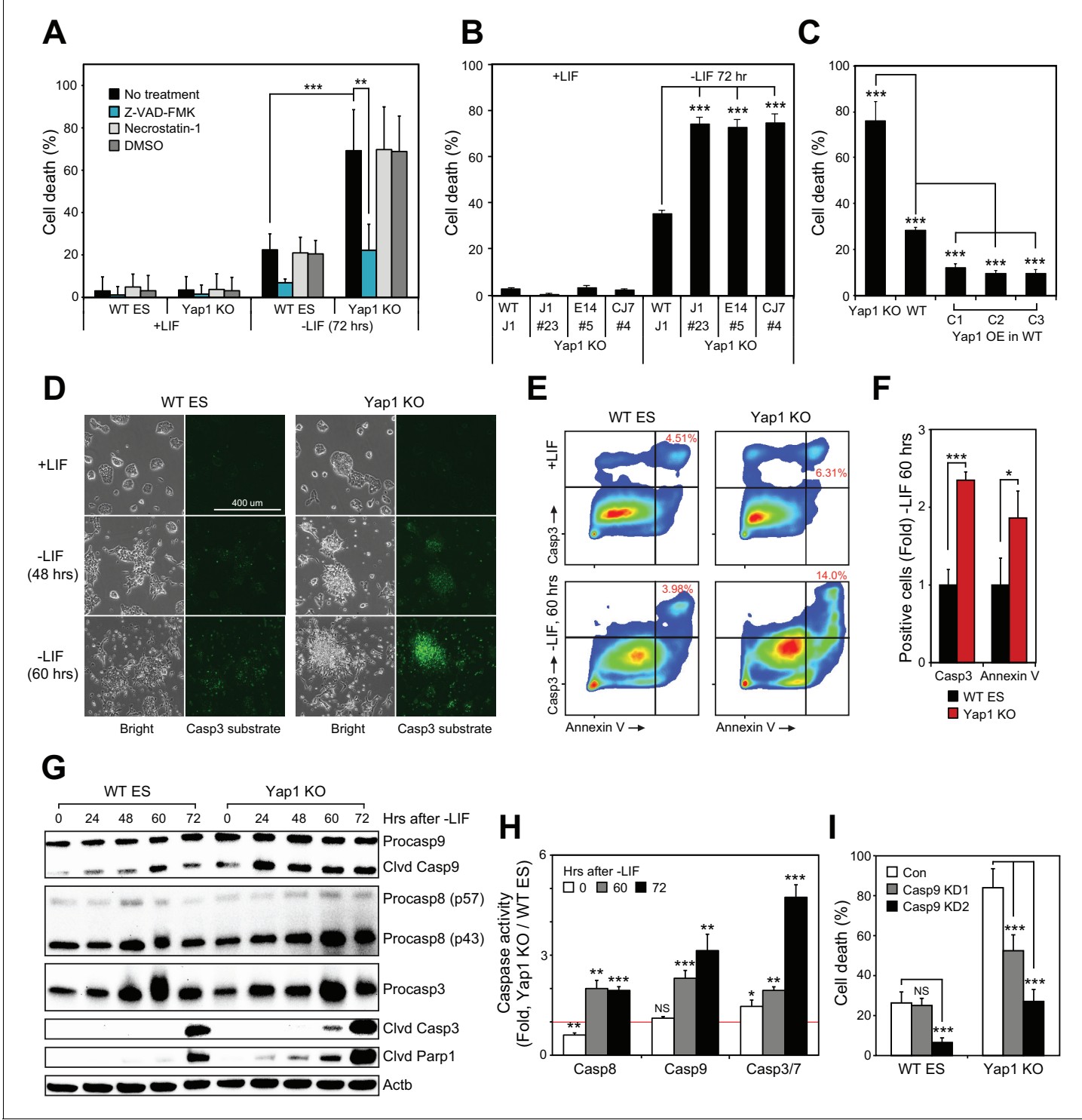

**Figure 1.** Loss of Yap1 substantially increases apoptosis during ESC differentiation. (A) Lactate dehydrogenase (LDH) assay of WT and Yap1 KO ESCs in ±LIF. Cells were treated with either Z-VAD-FMK (Z-VAD), necrostatin-1, DMSO, or no treatment. Values were normalized to wells that had been lysed completely. (B) LDH assay measuring cell death after Yap1 KO in three different ESC lines during differentiation (72 hr) or self-renewal. (C) LDH assay measuring cell death in Yap1 KO, WT, and three different stable FLAG-Bio (FB) Yap1 overexpression cell lines during differentiation (72 hr). (D) Representative brightfield and fluorescence microscopy images of WT and Yap1 KO ESCs incubated with NucView 488 Casp3 substrate at the indicated times after LIF withdrawal. (E) Representative flow cytometry density plots of WT and Yap1 KO ESCs detecting fluorescent signal from annexin-V (conjugated to CF594) and NucView 488 reagent during differentiation (60 hr). (F) Fold enrichment of annexin-V and active Casp3-positive Yap1 KO vs. WT ESCs according to flow cytometry. (G) Immunoblot of Casp9, Casp8, Casp3, cleaved Casp3, and cleaved Parp1 in WT and Yap1 KO

*Figure 1 continued on next page*

*Figure 1 continued*

cells during differentiation. β-actin was used as a loading control. (**H**) Luminescent assay of caspase activity in Yap1 KO vs. WT ESCs in ±LIF media. (**I**) LDH assay of WT and Yap1 KO cells ± KD of Casp9 during differentiation (72 hr). All data are expressed as mean ±standard deviation (n = 4 independent samples for LDH assays and n = 3 for other experiments). Two sample two-tailed t-test compared to WT or whatever is specified on the y-axis: *=0.05 > P > 0.01. **=0.01 > P > 0.001. ***=0.001 $\geq$ P.

DOI: https://doi.org/10.7554/eLife.40167.002

The following figure supplement is available for figure 1:

**Figure supplement 1.** Yap1 expression in KO/KD/OE cell lines, STS sensitivity, and caspase expression during ES cell differentiation.

DOI: https://doi.org/10.7554/eLife.40167.003

more intense cleavage of Casp9, Casp3, and Parp1 in Yap1 KO cells during differentiation (*Figure 1G*). To determine whether Yap1 KO dESCs were more sensitive to exogenous apoptosis-inducing stimuli, we treated dESCs with staurosporine (STS), a high-affinity, non-specific kinase inhibitor that has long been used to dissect the induction of intrinsic apoptosis in a myriad of cellular contexts (*Belmokhtar et al., 2001*; *Preta and Fadeel, 2012*; *Xu et al., 2015*). This treatment induced faster and more drastic activation of Casp3 and Parp1 in Yap1 KO than in WT dESCs as quickly as two hours after addition, reflecting a vastly heightened sensitivity to apoptosis-inducing stress (*Figure 1—figure supplement 1G*).

Next, we quantified caspase activity using a luminogenic substrate. By 60 hr after LIF removal, all caspases tested were approximately two-fold more active in Yap1 KO cells than in WT (*Figure 1H*). These observations demonstrate that lack of Yap1 accelerates and intensifies caspase activation during differentiation. We decided to dissect which part of the apoptotic pathway is affected first by loss of Yap1. Though Casp8 activity is elevated in Yap1 KO cells, we did not detect substantial differences in cell death after Casp8 KD (data not shown), so we decided to target Casp9 with two different shRNAs (*Figure 1—figure supplement 1H*). As expected, KD of Casp9 reduced cell death during differentiation, and this was particularly stark for Yap1 KO cells, where cell death was reduced to WT levels without Casp9 KD (*Figure 1I*). This implied that the abnormally high rates of apoptosis in Yap1 KO cells are sustained by heightened Casp9 activation. We observed that mRNA expression of caspases was relatively equal between Yap1 KO cells and WT cells during differentiation (*Figure 1—figure supplement 1I*). Additionally, protein levels of Casp3 showed similar fluctuations in dESCs for both WT and KO cells; although Casp8 and Casp9 were elevated in Yap1 KO cells (*Figure 1G*). However, since caspase activity is strongly activated by cleavage (*Hu et al., 2013*), we speculated that Yap1 may regulate other factors that indirectly affect the rate of caspase cleavage.

## Yap1 protects against apoptosis regardless of differentiation method and acts directly after the exit from self-renewal

To determine whether the roles of Yap1 are either specific to LIF withdrawal or broadly applicable to the exit from self-renewal in different conditions, we utilized alternate differentiation methods (*Figure 2A*). Utilizing N2B27 medium (neural ectoderm fate) or low serum DMEM supplemented with IDE1 (definitive endoderm fate) (*Borowiak et al., 2009*), we again observed that dESCs without Yap1 experienced much higher rates of cell death compared to WT cells, which could be rescued by zVAD (*Figure 2B and C*). We verified by RT-qPCR that N2B27 medium indeed induced neural ectoderm marker expression (*Figure 2—figure supplement 1A*) whereas IDE1 treatment induced endoderm marker expression (*Figure 2—figure supplement 1B*), as well as repression of *Nanog*, an ESC self-renewal marker.

We also induced differentiation towards epiblast-like cells (EpiLCs) to mimic early embryo development in vitro; whereas mESCs are equivalent to the inner cell mass of the blastocyst at E3.5–4.5, EpiLCs represent the next developmental stage, the E5.5–6.0 epiblast (*Hayashi et al., 2011*). We confirmed repression of *Nanog* and upregulation of EpiLC-specific markers (*Figure 2—figure supplement 1C*). As expected, EpiLCs lacking Yap1 underwent substantially higher cell death than WT by d3, and zVAD reduced cell death in both genotypes to the low, basal rates experienced in 2i media (*Figure 2D*). Finally, we used a well-characterized inhibitor of Yap1, verteporfin (*Brodowska et al., 2014*), to investigate Yap1's role during late -LIF differentiation (*Figure 2E*). While treatment with as low as 1 µM verteporfin before the exit from self-renewal phenocopied Yap1 KO, treatment during late differentiation had more modest effects on cell death, and treated

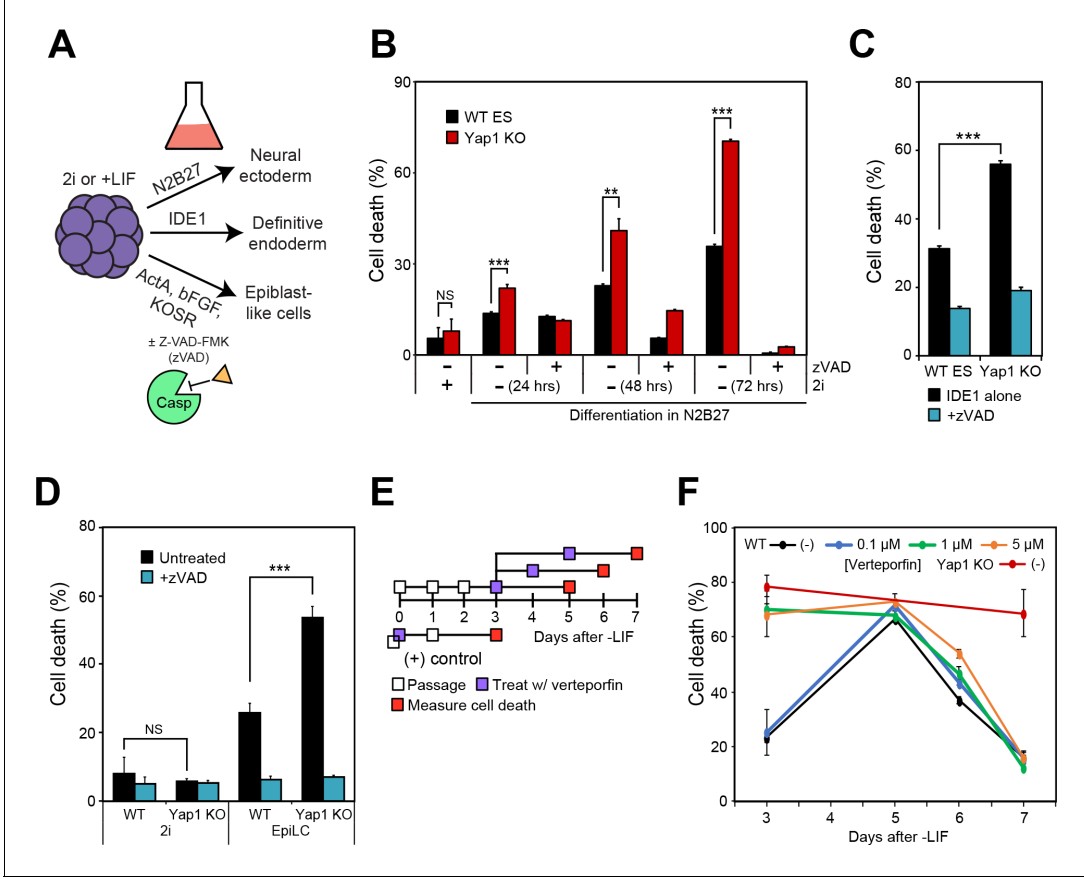

**Figure 2.** Loss of Yap1 augments apoptosis in several differentiation conditions, but its role is largely restricted to the exit from self-renewal. (**A**) Schematic of 3 differentiation protocols (ectoderm, endoderm, and epiblast) used in *Figures 2*, *3* and *5*. (**B**) LDH assay of WT and Yap1 KO ESCs in N2B27 with or without 2i and Z-VAD. (**C**) LDH assay of WT and Yap1 KO ESCs in low serum DMEM supplemented with IDE1 ±Z VAD (48 hr). (**D**) LDH assay of ESC towards EpiLC conversion in WT and Yap1 KO ESCs (72 hr). (**E**) Schematic of verteporfin (vert) treatment timings during late and early differentiation in WT ESCs in -LIF. (**F**) Timecourse LDH assay of verteporfin-treated dESCs at the indicated timepoints along with positive controls (treatment with verteporfin just after -LIF as well as untreated Yap1 KO ESCs, the latter of which are n = 8). All data are expressed as mean ±standard deviation (n = 4 independent samples unless otherwise stated). Two sample two-tailed t-test compared to WT or whatever is specified on the y-axis: *=0.05 > P > 0.01. **=0.01 > P > 0.001. ***=0.001 ≥ P.

DOI: https://doi.org/10.7554/eLife.40167.004

The following figure supplement is available for figure 2:

**Figure supplement 1.** Lineage marker expression during various differentiation methods.

DOI: https://doi.org/10.7554/eLife.40167.005

cells had death rates nearly identical to untreated by d7 (*Figure 2F*). Together, these data suggest that loss of *Yap1* increases rates of apoptosis in ESCs directly after the exit from self-renewal, regardless of the ultimate lineage those ESCs are destined for.

## Yap1 modulates the expression of apoptosis-related genes during differentiation

Following our deduction that Casp9 hyperactivation distinguishes Yap1 KO dESCs from WT dESCs and that lack of *Yap1* enhances apoptosis in several differentiation conditions, we examined the expression of anti- and pro-apoptotic genes that affect Casp9 activation. After 72 hr of LIF withdrawal, we detected a deficiency in three key anti-apoptotic proteins (Bcl-2, Bcl-xL, and Mcl-1) in Yap1 KO cells by immunoblot (*Figure 3A*). Immunocytochemistry confirmed reduced expression of Bcl-2 and Mcl-1 in Yap1 KO dESCs compared to WT, as well as lower mitochondrial content as measured by MitoTracker dye (*Figure 3—figure supplement 1A and B*); as expected, Bcl-2 and Mcl-1 strongly colocalized with the mitochondria (weighted colocalization coefficient for all samples ~ 0.7–

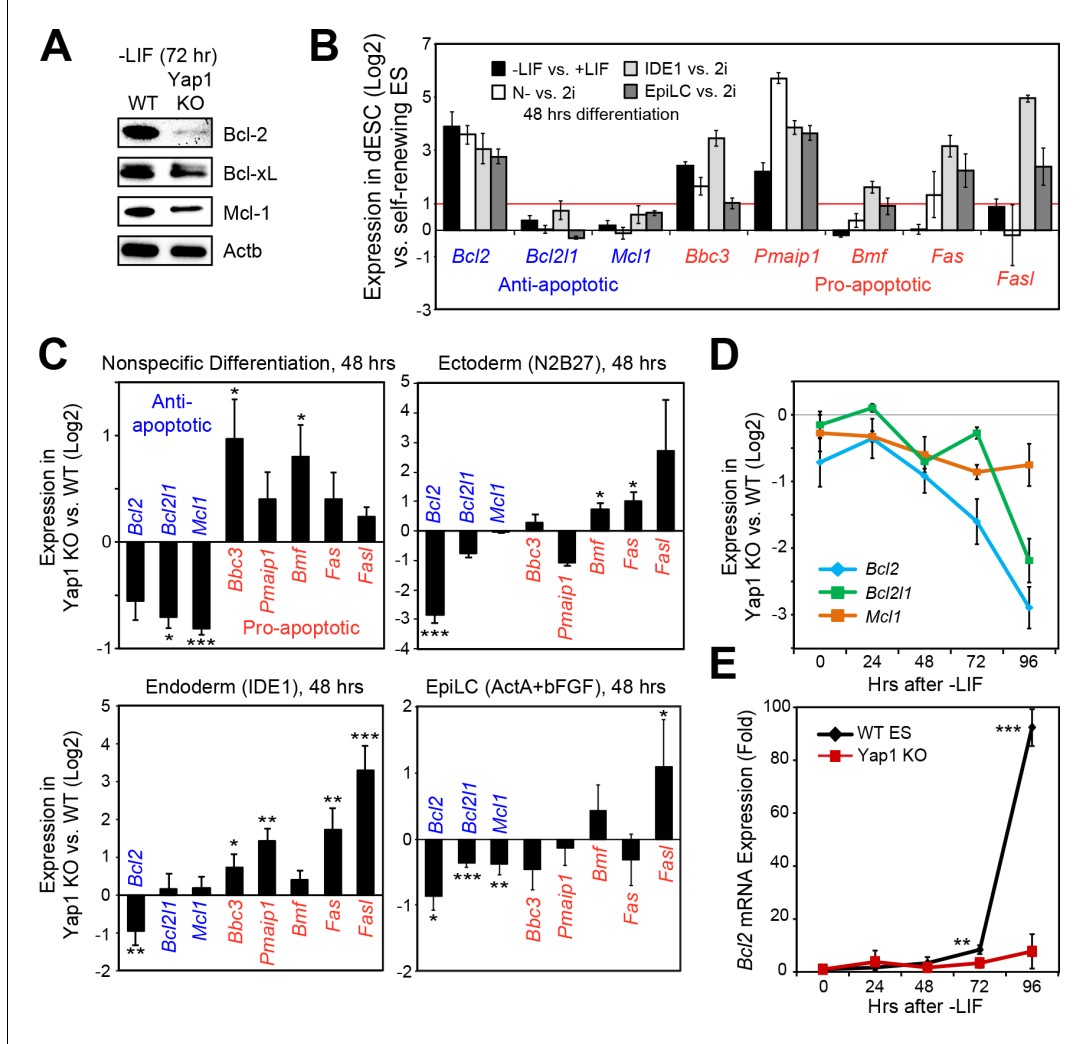

**Figure 3.** Loss of Yap1 leads to abnormal expression of apoptosis-related genes. (A) Immunoblot of Bcl-2, Bcl-xL, and Mcl-1 in WT and Yap1 KO cells in -LIF after 72 hr of differentiation. (B) RT-qPCR measuring the expression of anti-apoptotic (blue) and pro-apoptotic (red) genes in WT ESCs cultured in the indicated differentiation conditions (all at 48 hr) normalized to their respective self-renewal conditions. (C) RT-qPCR measuring the expression of anti- and pro-apoptotic genes in Yap1 KO vs. WT cells (log2) in various differentiation conditions (all at 48 hr). (D) RT-qPCR measuring the expression of *Bcl2*, *Bcl2l1*, and *Mcl1* in Yap1 KO cells vs. WT cells during differentiation (timecourse). (E) RT-qPCR measuring the expression of *Bcl2* in WT and Yap1 KO cells during differentiation (timecourse) relative to +LIF. All data are expressed as mean ±standard deviation (n = 3 independent samples unless otherwise stated). Two sample two-tailed t-test compared to WT or whatever is specified on the y-axis: *=0.05 > P > 0.01. **=0.01 > P > 0.001. ***=0.001 ≥ P.

DOI: https://doi.org/10.7554/eLife.40167.006

The following source data and figure supplement are available for figure 3:

**Source data 1.** Data used in *Figure 3—figure supplement 1D and E*.
DOI: https://doi.org/10.7554/eLife.40167.008
**Figure supplement 1.** Depletion or loss of Yap1 leads to dysregulation of apoptosis-related genes.
DOI: https://doi.org/10.7554/eLife.40167.007

0.9). We then investigated the significance of this expression defect in the context of what normally happens during differentiation. In WT ESCs, we found that *Bcl2* was strongly upregulated in all differentiation conditions tested; pro-apoptotic genes such as Puma (*Bbc3*) and Noxa (*Pmaip1*) were also activated, whereas *Bcl2l1* and *Mcl1* either stayed constant or were weakly upregulated (*Figure 3B*). Comparing Yap1 KO cells to WT ESCs, by d2, we found a general trend for decreased anti-apoptotic gene expression (most consistently *Bcl2*) and increased pro-apoptotic gene expression (*Figure 3C*). This defect worsened over time in -LIF (*Figure 3D*) and was particularly stark for

*Bcl2*, which was upregulated as much as 80 to 100-fold by 96 hr in WT ESCs upon differentiation (*Figure 3E*).

To reinforce this observation, we examined the expression of apoptosis-related genes after 2d of transient OE of Yap1 after 3d of differentiation total, and found a modest induction in *Bcl2*, *Bcl2l1*, and *Mcl1*, as well as a modest repression of *Bbc3* and *Bmf* (*Figure 3—figure supplement 1C*). Using RNA-seq data from a previous study (*Chung et al., 2016*), we found that differentiation induces the expression of a group of anti-apoptotic genes in WT cells, but this induction is debilitated after Yap1 KD (*Figure 3—figure supplement 1D*). Meanwhile, constitutive Yap1 OE during +LIF conditions appeared to slightly induce anti-apoptosis genes on average, though not significantly (*Figure 3—figure supplement 1E*). Collectively, these data show that Yap1 may function as a master regulator in proper maintenance or induction of anti-apoptotic genes (particularly *Bcl2*) during differentiation, and it may also dampen the upregulation of pro-apoptotic genes.

## Yap1 directly regulates apoptosis-related genes via transcription

We performed ChIP-seq of Yap1 using ESCs overexpressing FLAG-Bio-Yap1 (FB-Yap1) under differentiation (-LIF, 72 hr) and self-renewal (+LIF) conditions, detecting 8453 peaks significantly enriched over the BirA control above threshold between duplicates during differentiation and only 699 peaks in +LIF, reflecting its known cytoplasmic localization during self-renewal. Many of the differentiation-related peaks were intergenic as well as in promoters (*Figure 4—figure supplement 1A*). Yap1 occupancy was positively correlated with degree of gene downregulation upon Yap1 KD, although some upregulated genes upon KD were associated with unusually low Yap1 occupancy (*Figure 4A*). By integrating data from a previous study investigating enhancer patterns at different stages of pluripotency (*Buecker et al., 2014*), we found that Yap1 peaks were strongly correlated with increased Ep300 (p300) occupancy during differentiation (*Figure 4B*). Although EpiLC differentiation induces a different cell fate than -LIF due to supplementation with activin A and bFGF, we reasoned that apoptosis-related regulation would be shared between the two conditions. We confirmed a physical interaction between Yap1 and p300 as well as one of its known cofactors, Tead4 (*Chen et al., 2010*), during ESC differentiation using co-immunoprecipitation (*Figure 4—figure supplement 1B*), consistent with known Yap1 nuclear localization in dESCs (*Chung et al., 2016*). Indeed, motif analysis revealed a significant enrichment of the Tead factor motif in addition to Zic3 and AP-1 complex (JunB and Fra1 (Fosl1)) motifs in the center of Yap1 peaks, whereas Esrrb (a negative control) was not found (*Figure 4—figure supplement 1C and D*). Finally, since p300 possesses histone acetyltransferase activity, we confirmed an increase in H3K27ac, an activating histone mark, in Yap1 peaks during differentiation (*Figure 4—figure supplement 1E*). Gene ontology (GO) analysis of genes bound by Yap1 and downregulated by Yap1 KD mainly yielded terms related to cell migration and motility, and regulation of cell death was also statistically significant (*Figure 4—figure supplement 1F*).

In addition to its co-activating properties, Yap1 also acts as a co-repressor in other contexts (*Kim et al., 2015*), and we observed Yap1 occupancy on both anti-apoptotic and pro-apoptotic genes (*Figure 4C*). To characterize Yap1 target putative *cis*-regulatory elements, we performed the dual luciferase assay in Yap1 KO cells, WT cells, and cells transfected with a Yap1 OE vector using the pGL3 promoter vector (*Figure 4—figure supplement 1G*). In Yap1 KO cells, luciferase constructs with regulatory elements associated with *Mcl1*, *Bcl2*, or *Bcl2l1* have lower luciferase activity relative to WT cells, while regulatory elements associated with *Bmf*, *Pmaip1*, and *Bbc3* led to higher luciferase activity (*Figure 4D*). Meanwhile, transient OE of Yap1 led to higher luciferase activity with anti-apoptotic gene regulatory elements and lower luciferase activity with pro-apoptotic gene regulatory elements (*Figure 4E*). Though the initial *Bcl2* intronic regulatory element was unresponsive to Yap1 OE, combining it with another element in the same intron (*Figure 4—figure supplement 1H*) caused its activity to increase 2x during OE (*Figure 4E*).

To determine the importance of the known Yap1-Tead interaction for the function of these regulatory elements, we chose the strongest enhancers (*Mcl1* distal and *Bcl2* intronic tandem) for further testing. Transient OE of Yap1 in Yap1 KO cells rescued enhancer function to levels comparable to Yap1 OE in WT cells, whereas OE of Yap1 S79A, a mutant less capable of binding to Tead factors (*Schlegelmilch et al., 2011*), only mildly rescued *Mcl1*'s enhancer's activity and failed to rescue *Bcl2*'s enhancer's activity at all (*Figure 4F*). Furthermore, ablation of the Tead binding sequence (ΔTBS) from the *Mcl1* enhancer not only eliminated its Yap1 responsiveness, but also nearly

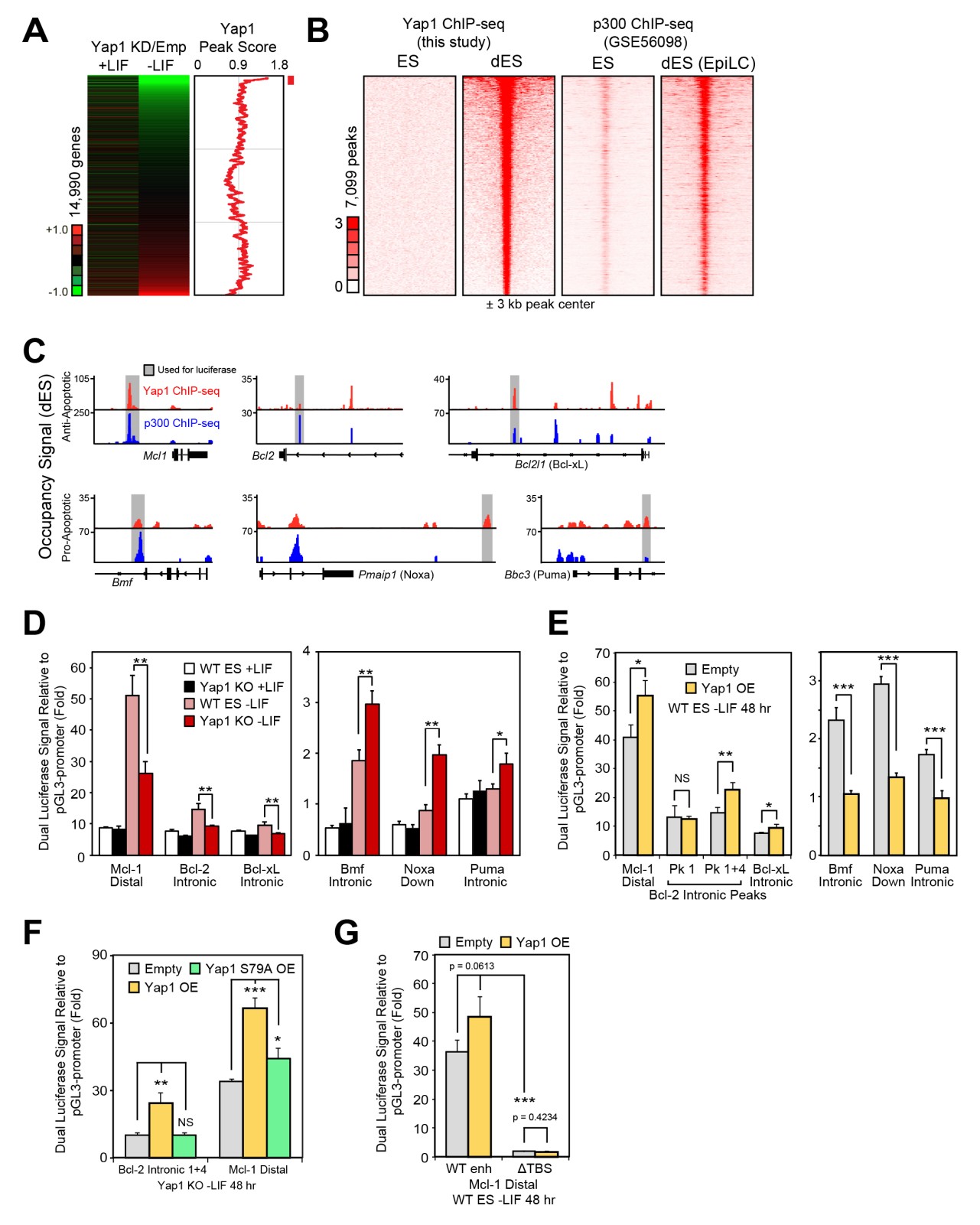

**Figure 4.** Yap1 directly regulates target apoptotic genes during differentiation. (**A**) RNA-seq heatmap (Yap1 KD/empty vector KD, in both undifferentiated and differentiating ESCs) and line graph depicting Yap1 peak score, normalized to BirA, calculated using a moving window average (window = 150). Color bar indicates extent of upregulation (red) or downregulation (green) upon Yap1 KD. (**B**) ChIP-seq peak heatmaps using coordinates centered on the top Yap1 peaks (p-value cutoff, 1e-5) in dESCs (-LIF), which are shown in the second heatmap from the left. The other

*Figure 4 continued*

heatmaps represent occupancy of Yap1 in ESCs (first) or p300 in ESCs (third) or dESCs (fourth) corresponding to Yap1 dESC peak centers ± 3 kb (bin size = 100). (C) Signal tracks of Yap1 (red) and p300 (blue) occupancy on apoptosis-related genes in dESCs and EpiLCs, respectively. (E and F) Dual luciferase assay of Yap1-occupied cis-regulatory elements from anti- and pro-apoptotic genes in (E) Yap1 KO and WT cells ± LIF (48 hr) or (F) WT cells with Yap1 or empty OE (in -LIF, 48 hr), relative to pGL3-promoter, 24 hr after transfection. (G) Dual luciferase assay of Bcl-2 and Mcl-1 regulatory elements in Yap1 KO cells after transfection of empty vector or vectors containing FLAG-Bio Yap1 with or without a Ser79Ala mutation. (H) Dual luciferase assay of Mcl-1 with a deletion of its Tead binding motif (GGAAT on the reverse strand) in WT ESCs ± Yap1 OE. All data are expressed as mean ±standard deviation (n = 3 independent samples unless otherwise stated). Two sample two-tailed t-test compared to WT or whatever is specified on the y-axis: *=0.05 > P > 0.01. **=0.01 > P > 0.001. ***=0.001 ≥ P.
DOI: https://doi.org/10.7554/eLife.40167.009

The following source data and figure supplement are available for figure 4:

**Source data 1.** Data used in *Figure 4A*, *Figure 4—figure supplement 1A,C,D,E,I and K*.
DOI: https://doi.org/10.7554/eLife.40167.011
**Figure supplement 1.** Yap1 binds to distal regulatory elements, primarily through Tead factors.
DOI: https://doi.org/10.7554/eLife.40167.010

abolished its enhancer activity (*Figure 4G*). Intriguingly, Yap1 occupancy on apoptosis-related genes seems to be relatively conserved (r =~0.5) among different human cancer cell types (*Figure 4—figure supplement 1I and J*). Thus, Yap1 may regulate apoptosis-related genes through conserved binding locations in both the human and mouse genome. Additionally, Yap1 peaks in mouse dESCs also correlated with Tead1 and Tead4 peaks in other mouse cell types, and signal tracks show similar occupancy patterns particularly for *Bcl2* (*Figure 4—figure supplement 1K and L*). Taken together, our data as well as data reanalyzed from other labs suggest that Yap1 directly regulates apoptosis-related genes.

## Loss of Yap1 contributes to heightened mitochondrial priming and dependence on anti-apoptotic proteins

Mitochondrial priming describes how close a cell is to the threshold of apoptosis and is a function of the balance between anti-apoptotic and pro-apoptotic proteins (*Czabotar et al., 2013*; *Deng, 2017*; *Sarosiek et al., 2013*). Since Yap1 KO cells already show higher expression of pro-apoptotic genes and lower expression of anti-apoptotic genes upon differentiation (*Figure 3C*), we surmised that loss of *Yap1* would increase mitochondrial priming and thereby sensitize dESCs to activation of the apoptotic cascade.

Using the JC-10 assay, we measured differences in mitochondrial priming between Yap1 KO and WT ESCs during differentiation, initially. Whereas mitochondria were equally primed during self-renewal, all four differentiation conditions (-LIF, neural, endoderm, EpiLC) resulted in a greater loss of mitochondrial membrane potential (Δψ) in Yap1 KO cells normalized to WT ESCs (*Figure 5A*). Next, we treated ESCs with a small panel of BH3 mimetics capable of inhibiting Bcl-2, Bcl-xL, Mcl-1, and/or Bcl-w to measure addiction to anti-apoptotic proteins. As expected, inhibition of anti-apoptotic proteins increased Δψ in dESCs more than in self-renewing ESCs (*Figure 5B*). Furthermore, deletion of *Yap1* significantly sensitized dESCs, but not undifferentiated ESCs, to Δψ loss post BH3 mimetic treatment. We then investigated whether the higher loss of Δψ in Yap1 KO cells correlated with greater rates of cell death. Our results showed that mere inhibition of anti-apoptotic proteins was sufficient to cause cell death, particularly in dESCs, even before apoptosis normally occurs during differentiation (*Figure 5C*). Strikingly, loss of *Yap1* significantly amplified cell death in response to BH3 mimetics at almost all concentrations tested, but only during differentiation (*Figure 5C*). Ablation of *Yap1* also enhanced addiction to Mcl-1 and Bcl-xL; inhibition of either protein resulted in 2-3x greater cell death in Yap1 KO than WT (*Figure 5C*). Thus, loss of *Yap1* leads to increased mitochondrial priming during differentiation, which subsequently sensitizes Yap1 KO to excessive activation of the apoptotic cascade. Importantly, despite how critical mitochondrial priming is to biomedical applications such as successful chemotherapy, almost no genes that regulate mitochondrial priming upstream of apoptosis-related proteins have been shown in any context.

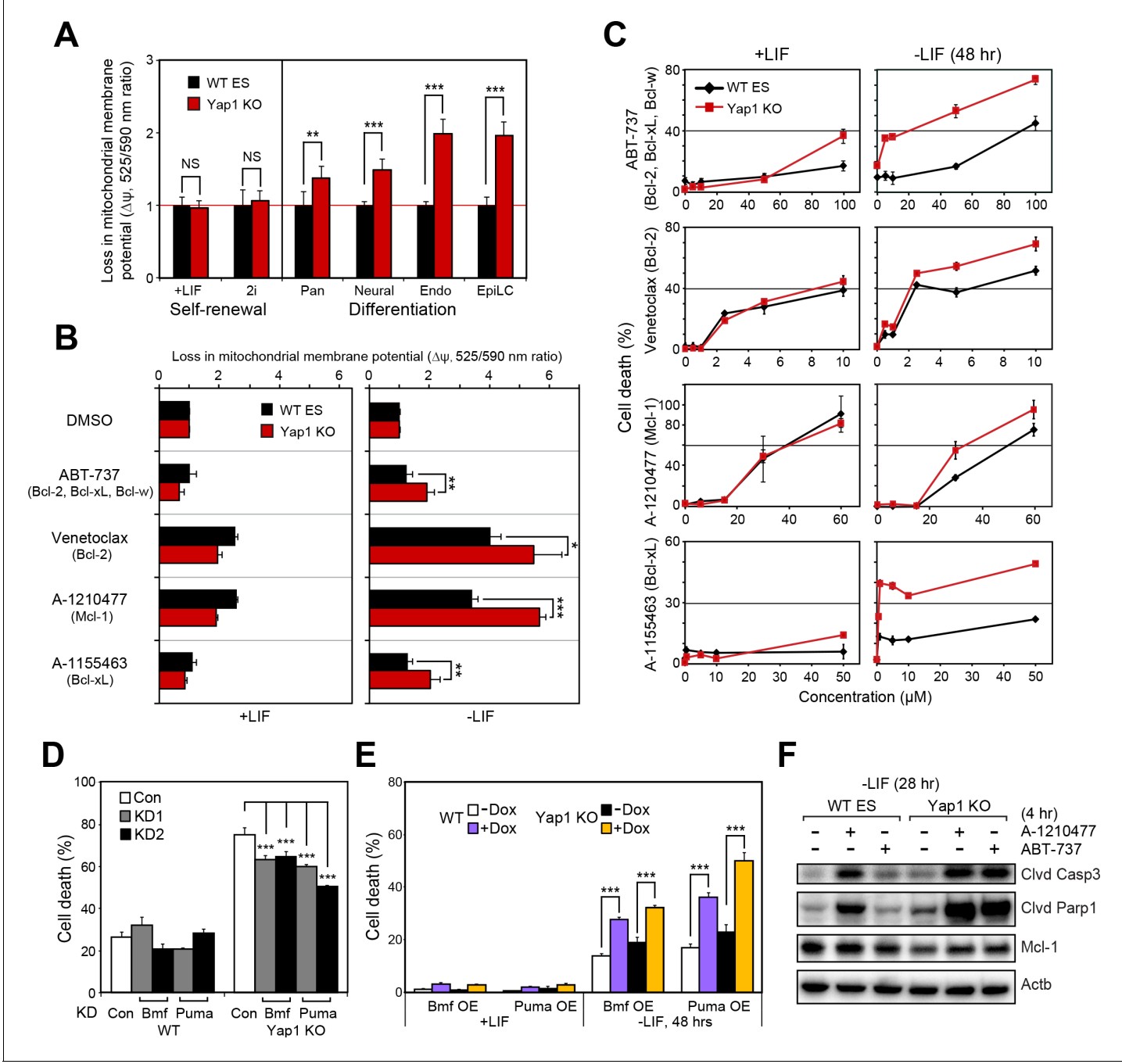

**Figure 5.** Yap1 regulates mitochondrial priming and addiction to anti-apoptotic proteins. (A) JC-10 mitochondrial membrane potential assay in WT and Yap1 KO cells during various forms of differentiation (72 hr for Pan and EpiLC, 48 hr for Neural and Endo) and self-renewal (maintained for an equal amount of time). Values (525/570 nm ratio, n = 6) corresponding to loss in Δψ (mitochondrial membrane potential) in Yap1 KO cells were normalized to WT cells. (A) JC-10 assay in WT and Yap1 KO cells in ±LIF after 12 hr of treatment with BH3 mimetics ABT-737, Venetoclax, A-1210477, and A1155463 (total differentiation time: 36 hr). Values (525/570 nm ratio) corresponding to loss in Δψ were normalized to DMSO as a control. (C) LDH assays of BH3 mimetic dose response curves after 24 hr of treatment in WT and Yap1 KO cells in ±LIF (48 hr differentiation). (D) LDH assay of WT and Yap1 KO cells after KD of Bmf or Puma in -LIF conditions (72 hr). (E) LDH assay of inducible Bmf and Puma OE (±Dox, 48 hr, 500 ng/mL) in WT and Yap1 KO cells in ±LIF (48 hr differentiation). (F) Immunoblot of cleaved Casp3, cleaved Parp1, and Mcl-1 in WT and Yap1 KO dESCs (28 hr) after 4 hr of treatment with BH3 mimetics A-1210477 (Mcl-1 inhibitor) and ABT-737 (inhibitor of Bcl-2, Bcl-xL, and Bcl-w). β-actin was used as a loading control. All data are expressed as mean ±standard deviation (n = 4 independent samples unless otherwise stated). Two sample two-tailed t-test compared to WT or whatever is specified on the y-axis: *=0.05 > P > 0.01. **=0.01 > P > 0.001. ***=0.001 ≥ P.

DOI: https://doi.org/10.7554/eLife.40167.012

*Figure 5 continued on next page*

*Figure 5 continued*

The following figure supplement is available for figure 5:

**Figure supplement 1.** Verification of KD and OE of pro-apoptotic factors Bmf and Puma.

DOI: https://doi.org/10.7554/eLife.40167.013

## Manipulation of the levels of BH3-only proteins Bmf or Puma mildly affect rates of ESC death in the absence of Yap1 during differentiation

BH3 mimetics promote cell death by mimicking pro-apoptotic BH3-only proteins (*Dai et al., 2016*). In addition to lower expression of anti-apoptotic proteins, we observed higher expression of BH3-only genes in Yap1 KO cells relative to WT cells (*Figure 3C*). Though it is known that KD of Puma in self-renewing ESCs reduces sensitivity to cytotoxic agents (*Huskey et al., 2015*), roles of Bmf and Puma during differentiation are relatively unknown. Therefore, we performed KD of Bmf and Puma (*Figure 5—figure supplement 1A and B*) and this mildly reduced cell death in Yap1 KO cells during differentiation but not in WT ESCs (*Figure 5D*). Inducible OE of either factor (*Figure 5—figure supplement 1C and D*) accelerated apoptosis during differentiation, particularly in Yap1 KO cells, and Puma promoted cell death more strongly than Bmf (*Figure 5E*). This difference may be because Puma promiscuously binds to all known anti-apoptotic Bcl-2 family proteins, whereas Bmf binds only weakly to Mcl-1, preferring Bcl-2, Bcl-xL, and Bcl-w (*Chen et al., 2005*). Thus, although Yap1's pro-survival function is primarily via activation of anti-apoptotic proteins, heightened expression of individual pro-apoptotic BH3-only proteins seems to contribute to enhanced cell death during differentiation of Yap1 KO cells. Finally, we used BH3 mimetics to probe differential roles of anti-apoptotic proteins during early differentiation. Mcl-1 expression was already reduced in Yap1 KO cells 28 hr after LIF withdrawal compared to WT ESCs. Yap1 KO cells were acutely sensitive to inhibition (4 hr) of either Mcl-1 or Bcl-2/Bcl-xL/Bcl-w, indicating increased mitochondrial priming even at such an early timepoint (*Figure 5F*). Since Mcl-1 is much more highly expressed than both Bcl-2 and Bcl-xL, we surmise that deficiency in its expression helps explain heightened apoptotic activation in Yap1 KO cells even before differences in Bcl-2 expression become apparent (*Figure 3E*).

## Modulation of anti-apoptotic proteins controls cell death during differentiation

Having shown that Yap1 directly regulates apoptosis-related genes and thus reduces mitochondrial priming during differentiation, we sought to characterize whether modulating individual Yap1 targets could control cell death during differentiation. We stably overexpressed Yap1 (as a positive control to complement the KO) and Bcl-xL (*Figure 6—figure supplement 1A*) and inducibly overexpressed Bcl-2 (*Figure 6—figure supplement 1B, C and D*) in Yap1 KO cells, which reduced cell death in Yap1 KO to levels comparable to WT (*Figure 6A and B*). Intriguingly, inducible OE of Taz, a Yap1 paralog also possessing a Tead-binding domain, reduced cell death in both Yap1 KO cells and WT cells to levels just below uninduced WT cells, perhaps via upregulation of Bcl-xL (*Figure 6C and D*), and *Figure 6—figure supplement 1F*). Conversely, KD of Bcl-xL, Mcl-1 (*Figure 6—figure supplement 1F*), or Bcl-2 (*Figure 6—figure supplement 1G*) in WT ESCs individually increased cell death during differentiation 1.5 to 2-fold compared to controls (*Figure 6E and F*, *Figure 6—figure supplement 1H*). Since Yap1 is crucial for ES differentiation, we questioned whether the apoptosis-related genes regulated by Yap1 might have some effect on differentiation efficiency. Surprisingly, we found that OE of Bcl-2 led to increased induction of trophectoderm (*Cdx2* and *Gata3*) and mesoderm markers (*Gsc* and *T*), while KD of Bcl-2 tended to reduce induction of lineage markers. However, KD of Bcl-xL or Mcl-1 had no effect on lineage marker induction (*Figure 6G* and *Figure 6—figure supplement 1I and J*). Taken together, these data clearly demonstrate that anti-apoptotic factors transcriptionally regulated by Yap1 are critical for dESC survival, and that OE or KD of each apoptotic factor can significantly shift the balance between survival and death. The results additionally suggest the previously unknown roles of Bcl-2 in regulation of ESC lineage specification, as only its roles in self-renewal have been deeply probed (*Yamane et al., 2005*). Our combined model of Yap1's role in ESC differentiation is provided in *Figure 6H*.

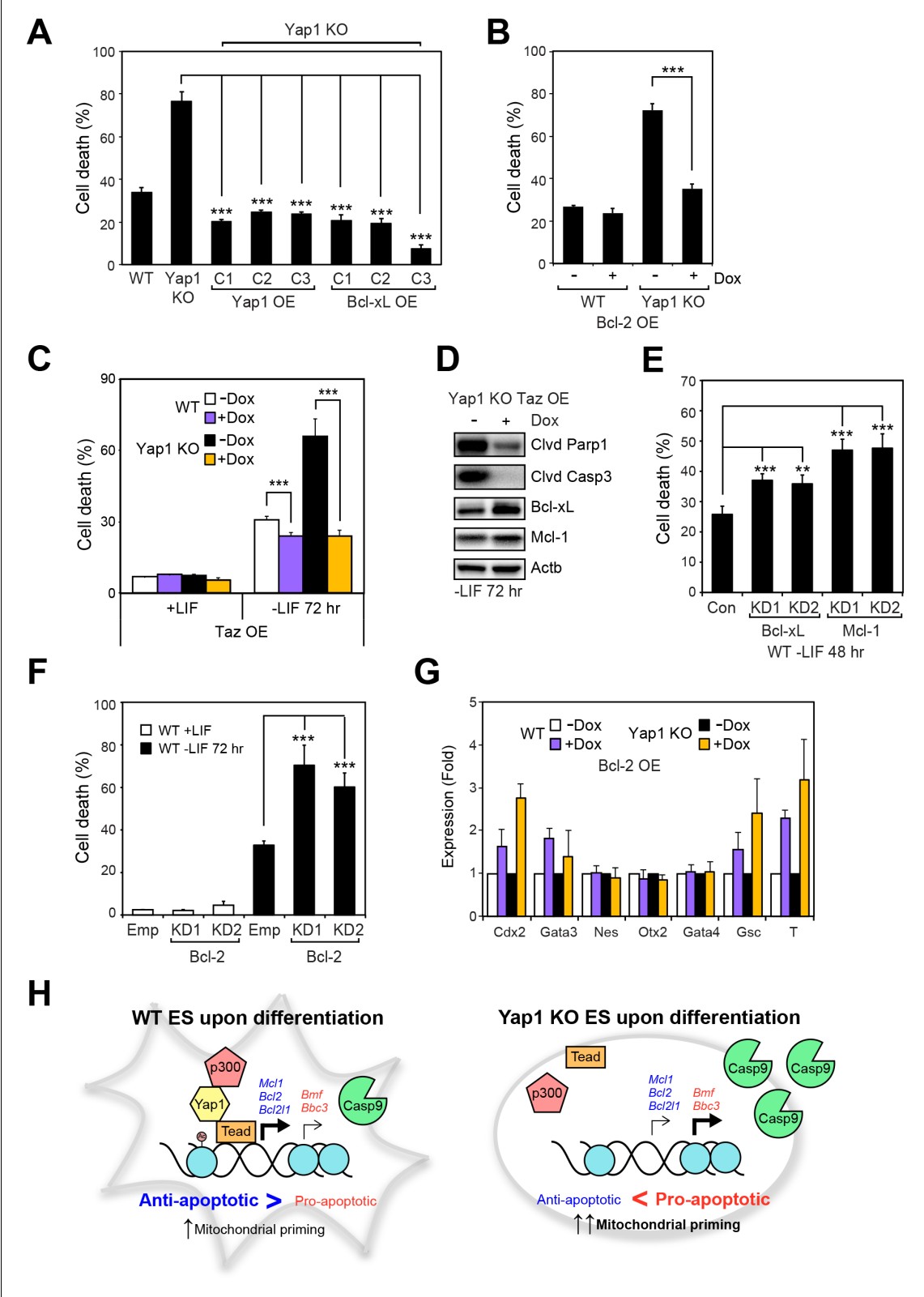

**Figure 6.** Overexpression of Taz or individual anti-apoptotic proteins fully rescues the survival defect in the absence of Yap1. (**A**) LDH assay of WT, Yap1 KO, and Yap1 KO constitutively overexpressing Bcl-xL or Yap1 in -LIF (72 hr). (**B**) LDH assay of inducible Bcl-2 (±Dox, 48 hr, 500 ng/mL) in WT and Yap1 KO cells -LIF (72 hr). (**C**) LDH assay of inducible Taz (±Dox, 48 hr, 500 ng/mL) in WT and Yap1 KO cells ± LIF (72 hr differentiation). (**D**) Immunoblot of cleaved Parp1, cleaved Casp3, Bcl-xL, and Mcl-1 in Yap1 KO cells inducibly overexpressing Taz (±Dox, 48 hr, 500 ng/mL) in -LIF (72 hr). (**E**) LDH assay

*Figure 6 continued on next page*

*Figure 6 continued*

of WT ESCs during differentiation (72 hr) after 48 hr KD of Bcl-xL or Mcl-1. (F) LDH assay of WT ESCs ± LIF (72 hr)±KD of Bcl-2. (G) RT-qPCR measuring the expression of lineage markers (trophectoderm: *Cdx2* and *Gata3*, ectoderm: *Nes* and *Otx2*, endoderm: *Gata4*, mesoderm: *Gsc* and *T*) in WT and Yap1 KO cells in -LIF (72 hr, n = 3). Expression is indicated as a fold change in +Dox samples relative to -Dox. (H) Model proposing roles for Yap1 specific to the exit from self-renewal. In complex with Tead factors like Tead4, Yap1 co-activates anti-apoptotic genes and mildly co-represses pro-apoptotic genes to dampen mitochondrial priming, which thus prevents hyperactivation of the apoptotic cascade through Casp9. All data are expressed as mean ±standard deviation (n = 4 independent samples unless otherwise stated). Two sample two-tailed t-test compared to WT or whatever is specified on the y-axis: *=0.05 > P > 0.01. **=0.01 > P > 0.001. ***=0.001 ≥ P.

DOI: https://doi.org/10.7554/eLife.40167.014

The following figure supplement is available for figure 6:

**Figure supplement 1.** Modulation of the expression of individual anti- or pro-apoptotic genes influences cell death during differentiation.
DOI: https://doi.org/10.7554/eLife.40167.015

## Discussion

Though ESCs experience 30% or more cell death during differentiation, regulators of this process remain largely unknown. In this study, we have shown that in the absence of *Yap1*, this proportion of cell death increases to 70–80%. Accordingly, we have demonstrated that Yap1 directly and strongly activates anti-apoptotic genes, in addition to mildly repressing pro-apoptotic genes, to promote survival during the stressful process of differentiation. Yap1 therefore attenuates the increase in mitochondrial priming during differentiation that threatens mitochondrial integrity and leads to Casp9 activation. OE of Yap1, its paralog Taz, or its anti-apoptotic targets in Yap1 KO cells reduces cell death to WT levels or even lower. Our proposed role for Yap1 as a pro-survival factor in ESCs is consistent with other studies done in cancer or epithelial contexts (*Lin et al., 2015*; *Rosenbluh et al., 2012*; *Song et al., 2015*; *Zhao et al., 2016*), but our work is the first study to show such a contextual, differentiation-specific role for Yap1 in ESCs.

Intriguingly, our Casp3 live imaging assay revealed that activation of Casp3 was extremely heterogeneous, with many cells changing their morphology during differentiation without detectable caspase activation. Future studies focusing on how individual cells make the molecular decision of differentiation vs. apoptosis will be desired. We hypothesize that relative changes in the expression of key pro- and anti-apoptotic genes at the single cell level, as well as lineage markers, shortly after LIF withdrawal could successfully predict whether an individual cell will differentiate or perish.

One unexpected finding from our study is that OE of Bcl-2 improved induction of essential trophectoderm and mesoderm markers, and KD of Bcl-2 (but not Mcl-1 or Bcl-xL) conversely hampered such induction. Elucidating the mechanism by which Bcl-2 accelerates induction of lineage markers is beyond the scope of this work but would enhance understanding of how apoptosis-related factors influence non-apoptotic processes such as differentiation. Additionally, we noted that Yap1 binding peaks on apoptosis-related genes are conserved across several human cancer cell types, which corroborates previous findings (*Rosenbluh et al., 2012*). Since addiction to anti-apoptotic factors is a defining characteristic of cancer cells, the regulation mechanisms we have elucidated may be broadly applicable to how Yap1 promotes tumorigenesis.

In sum, our study has clearly demonstrated that Yap1 robustly promotes survival of ESCs during differentiation by direct transcriptional regulation of apoptotic genes. Nearly all cells in the body originate from various progenitor cells, and since the process of differentiation is often fraught with error and stress, our research may spur advances in the regulation of the survival or death decision during cell fate changes in a broad variety of contexts.

## Materials and methods

**Key resources table**

| Reagent type (species) or resource | Designation | Source or reference | Identifiers | Additional information |
| --- | --- | --- | --- | --- |

*Continued on next page*

*Continued*

| Reagent type (species) or resource | Designation | Source or reference | Identifiers | Additional information |
|---|---|---|---|---|
| Antibody | Mouse anti-β-actin | Abgent | Cat#AM1829b RRID:AB_10664137 | 1:20,000 in 5% BSA |
| Antibody | Mouse anti-Yap1 | Santa Cruz Biotechnology | Cat#sc-101199 RRID:AB_1131430 | 1:1000 in 5% milk |
| Antibody | Rabbit anti-cleaved caspase-3 | Cell Signaling Technology | Cat#9661S RRID:AB_2341188 | 1:1000 in 5% BSA |
| Antibody | Mouse anti-cleaved Parp1 | Cell Signaling Technology | Cat#9548S RRID:AB_2160592 | 1:1000 in 5% BSA |
| Antibody | Mouse anti-caspase-9 | Cell Signaling Technology | Cat#9508S RRID:AB_10695598 | 1:1000 in 5% BSA |
| Antibody | Rabbit anti-Bcl-2 | Cell Signaling Technology | Cat#3498S RRID:AB_1903907 | 1:1000 in 5% BSA (WB), 1:200 in 4% BSA, 1% NGS (ICC) |
| Antibody | Rabbit anti-Bcl-xL | Cell Signaling Technology | Cat#2764S RRID:AB_10695729 | 1:1000 in 5% BSA |
| Antibody | Rabbit anti-Mcl-1 | Cell Signaling Technology | Cat#94296S RRID:AB_2722740 | 1:1000 in 5% BSA, 1:800 in 4% BSA, 1% NGS (ICC) |
| Antibody | Mouse anti-Puma | Santa Cruz Biotechnology | Cat#sc-374223 RRID:AB_10987708 | 1:500 in 5% BSA |
| Antibody | Rabbit anti-Bmf | Bioss | Cat#bs-7587R RRID:AB_2722741 | 1:1000 in 5% BSA |
| Antibody | Horse anti-mouse secondary, HRP-conjugated | Cell Signaling Technology | Cat#7076P2 RRID:AB_330924 | 1:10,000 in TBST |
| Antibody | Goat anti-rabbit secondary, HRP-conjugated | Cell Signaling Technology | Cat#7074S RRID:AB_2099233 | 1:10,000 in TBST |
| Antibody | Goat anti-rabbit IgG Alexa Fluor 594 | Thermo Scientific | Cat#R37117 RRID:AB_2556545 | 1:1000 in 4% BSA, 1% NGS (ICC) |
| Antibody | Dynabeads My One Streptavidin T1 | Thermo Scientific | Cat#65601 | 1:2000 in 5% BSA |
| Antibody | Rabbit anti-Taz (Wwtr1) | Sigma Aldrich | HPA007415 RRID:AB_1080602 | 1:500 in 5% milk |
| Chemical compound, drug | Z-VAD-FMK | ApexBio | Cat#A1902 | |
| Chemical compound, drug | Necrostatin-1 | Selleck Chemicals | Cat#S8037 | |
| Chemical compound, drug | CHIR99021 | Selleck Chemicals | Cat#S2924 | |
| Chemical compound, drug | PD184352 | Selleck Chemicals | Cat#S1020 | |
| Chemical compound, drug | IDE1 | Cayman Chemical Company | Cat#13816 | |

*Continued on next page*

*Continued*

| Reagent type (species) or resource | Designation | Source or reference | Identifiers | Additional information |
|---|---|---|---|---|
| Chemical compound, drug | Staurosporine | Cell Signaling Technology | Cat#9953S | |
| Chemical compound, drug | Polybrene | Millipore | Cat#TR-1003-G | |
| Chemical compound, drug | Puromycin | Thermo Scientific | Cat#A111 3803-02 | |
| Chemical compound, drug | Geneticin /G418 | Thermo Scientific | Cat#10131027 | |
| Chemical compound, drug | ABT-737 | Selleckchem | Cat#S1002 | |
| Chemical compound, drug | Venetoclax /ABT-199 | Selleckchem | Cat#S8048 | |
| Chemical compound, drug | A-1210477 | Selleckchem | Cat#S7790 | |
| Chemical compound, drug | A-1155463 | Selleckchem | Cat#S7800 | |
| Chemical compound, drug | Lipofectamine 3000 | Life Technologies | Cat#L3000008 | |
| Chemical compound, drug | INTERFERin | Polyplus Transfection | Cat#409–10 | |
| Chemical compound, drug | Verteporfin | Selleck Chemicals | S1786 | |
| Chemical compound, drug | Recombinant Human /Mouse/Rat Activin A Protein | R and D Systems | 338-AC-010 | |
| Chemical compound, drug | Gibco FGF Basic Recombinant Mouse Protein | Thermo Fisher Scientific | PMG0034 | |
| Chemical compound, drug | KnockOut Serum Replacement | Thermo Fisher Scientific | 10828028 | |
| Commercial assay or kit | Pierce LDH Cytotoxicity Assay Kit | Thermo Scientific | Cat#88954 | |
| Commercial assay or kit | RNeasy Plus Mini Kit | Qiagen | Cat#74136 | |
| Commercial assay or kit | qScript cDNA SuperMix | QuantaBio /VWR | Cat#101414–108 | |
| Commercial assay or kit | PerfeCTa SYBR Green FastMix | VWR | Cat#95072–012 | |
| Commercial assay or kit | Caspase-Glo 3/7 | Promega | Cat#G8090 | |

*Continued on next page*

*Continued*

| Reagent type (species) or resource | Designation | Source or reference | Identifiers | Additional information |
|---|---|---|---|---|
| Commercial assay or kit | Caspase-Glo 8 | Promega | Cat#G8200 | |
| Commercial assay or kit | Caspase-Glo 9 | Promega | Cat#G8210 | |
| Commercial assay or kit | Cell Meter JC-10 Mitochondrion Membrane Potential Assay Kit | AAT Bioquest | Cat#22800 | |
| Commercial assay or kit | NEBNext Ultra II DNA Library Prep Kit for Illumina | New England Biolabs | Cat#E7645S | |
| Commercial assay or kit | Dual-Glo Luciferase Assay System | Promega | Cat#E2920 | |
| Recombinant DNA reagent | pLKO-puro | Millipore Sigma | See table S3 | |
| Recombinant DNA reagent | pLVX-TRE3G -ZsGreen1 | Clontech | Cat#631164 | |
| Recombinant DNA reagent | pCMV-Tet3G | Clontech | Cat#631164 | |
| Recombinant DNA reagent | pCMV3-Bcl2l1/Bcl-xL | Sino Biological | Cat#MG50012-UT | |
| Recombinant DNA reagent | pGL3-promoter | Promega | Cat#E1761 | |
| Recombinant DNA reagent | pRL-TK | Promega | Cat#E2231 | |
| Recombinant DNA reagent | 3149 pSFFV-neo Bcl-2 cDNA | AddGene | Cat#8750 | |
| Recombinant DNA reagent | Mus musculus BCL2 binding component 3 (Bbc3), mRNA. NM_133234.2 | GenScript | Cat#OMu19350D | |
| Recombinant DNA reagent | pcDNA3.1/HisC-mTAZ | AddGene | Cat#31793 | |
| Cell line (*Mus musculus*, male) | J1 Embryonic Stem Cells | ATCC | ATCC SCRC-1010 | |
| Cell line (*M. musculus*, male) | CJ7 Embryonic Stem Cells | ENCODE | RRID:CVCL_C316 | |
| Cell line (*M. musculus*, male) | ES-E14TG2a Embryonic Stem Cells | ATCC | ATCC CRL-1821 | |
| Cell line (*Homo sapiens*) | HEK293T cells | ATCC | ATCC CRL-3216 | |
| Software, algorithm | FlowJo | Treestar | | |
| Software, algorithm | BoxPlotR | http://shiny.chemgrid.org/boxplotr/ | | |

*Continued*

| Reagent type (species) or resource | Designation | Source or reference | Identifiers | Additional information |
|---|---|---|---|---|
| Software, algorithm | Java TreeView | http://jtreeview.sourceforge.net/ | | |
| Software, algorithm | AmiGO 2 | http://amigo.geneontology.org | | |
| Software, algorithm | Primer3 | http://primer3.ut.ee/ | | |
| Software, algorithm | HOMER | http://homer.ucsd.edu/homer/ | | |
| Software, algorithm | GOrilla | http://cbl-gorilla.cs.technion.ac.il/ | | |
| Software, algorithm | Cistrome | http://cistrome.org/db/#/ | | |
| Software, algorithm | Galaxy | http://cistrome.org/ap/ | | |
| Software, algorithm | SRA Toolkit | https://trace.ncbi.nlm.nih.gov/Traces/sra/sra.cgi?view=toolkit_doc | | |
| Software, algorithm | Bowtie 2 | http://bowtie-bio.sourceforge.net/bowtie2/index.shtml | | |
| Software, algorithm | MACS2 | https://github.com/taoliu/MACS | | |
| Software, algorithm | Vassar Stats | http://vassarstats.net/matrix2.html | | |
| Software, algorithm | Integrated Genome Viewer | http://software.broadinstitute.org/software/igv/ | | |
| Software, algorithm | ZEN Microscope Software | https://www.zeiss.com/microscopy/int/downloads/zen.html | | |
| Software, algorithm | ImageJ | https://imagej.nih.gov/ij/index.html | | |
| Software, algorithm | STAR | https://github.com/alexdobin/STAR | | |

## Cell culture

J1, CJ7, and E14TG2a (E14) male mouse ESCs were cultured on 0.1% gelatin-coated plates in Dulbecco's Modified Eagle's Medium (DMEM, Gibco) supplemented with 18% fetal bovine serum (Bio-West), MEM nonessential amino acids (Gibco), EmbryoMax nucleosides (Millipore), 50 U/mL penicillin/streptomycin/L-glutamine (PSG, Gibco), 100 μM β-mercaptoethanol, and 1000 U/mL recombinant mouse leukemia inhibitory factor (LIF, Millipore) in a 37°C with 5% $CO_2$. Media was changed daily, and cells were passaged every 2 days. J1, CJ7, E14TG2a mouse ES cells and HEK293T cells were all obtained from ATCC (except CJ7 line was obtained from Dr. Stuart Orkin), confirmed by partial genomic DNA sequencing. Mycoplasma contamination was not detected by PCR based methods.

## Cell death assay

ESCs were seeded at a density of $2 \times 10^5$ cells/mL in the indicated media type in a clear 96-well plate. Inhibitors were administered 24 hr after seeding at a concentration of 50 μM (Z-VAD-FMK, ApexBio; Necrostatin-1, Selleckchem). At the indicated timepoints, lactate dehydrogenase (LDH) activity was quantified in the supernatant using the Pierce LDH Cytotoxicity Assay Kit (Thermo Scientific) according to the manufacturer's instructions. $A_{680nm}$ values were first subtracted as background

noise. Then, absorbance from an average of 3 media-only wells (reflecting background LDH activity) was subtracted from every sample's $A_{490nm}$ value. Data were normalized to wells that had been lysed completely using the provided lysis buffer to establish a benchmark for 100% cell death.

## Cell differentiation

For nonspecific differentiation, ESCs were washed in -LIF medium and then seeded at the cell densities specified below in each assay, as well as passaged on day one after seeding. Cells were assayed at 24, 48, 60, and/or 72 hr according to the experiment. For neural differentiation, ESC were first maintained in NDiff 227 medium (N2B27, Clontech) supplemented with 3 µM CHIR99021 (Selleck Chemicals) and 1 µM PD184352 (Selleck Chemicals), defined as 2i, to promote self-renewal in the absence of serum and LIF. Differentiation occurred in N2B27 in the absence of 2i and assayed at 24, 48, and 72 hr. For definitive endoderm differentiation, ESCs were grown in DMEM supplemented with 1% FBS, PSG, and 5 µM IDE1 (Cayman Chemical Company) and assayed at 48 hr. For EpiLC differentiation, ESCs were grown in N2B27 supplemented with 20 ng/mL activin A (R and D Systems), 12 ng/mL bFGF (Thermo Fisher), and 1% KOSR (Thermo Fisher) as previously shown (*Hayashi et al., 2011*), and assayed at 72 hr. For verteporfin-related experiments, verteporfin was diluted in DMSO to a concentration of 1 mM and then further diluted in fresh media during media changes, and cells were protected from light with aluminum foil. HEK293T cells (ATCC CRL-3216) were cultured in DMEM supplemented with 10% FBS and PSG. All cells were grown at 37°C in the presence of 5% $CO_2$.

## Flow cytometry

For analysis of externalized phosphatidylserine and active caspase-3, ESCs were seeded at a density of $4 \times 10^5$ cells per well on a six well plate in medium with or without LIF. After incubating cells for the indicated number of hours, cells were gently detached and dissociated into a single cell suspension using Accutase (Biolegend). Cells were then resuspended in 200 µL 1X Annexin V binding buffer plus 5 µL of 0.2 mM NucView 488 caspase-3 substrate solution and 5 µL CF 594 annexin V solution (Biotium). Cells were incubated in the dark at room temperature for 30 mins, then centrifuged (1000 rpm, five mins) at 4°C and washed. Then, stained cells were filtered using a 70 µm cell strainer to remove clumps (Celltreat). Flow cytometry was performed on a BD LSRFortessa SORP Flow Cytometer (BD Biosciences) and analysis was carried out with FlowJo (Treestar).

## Immunoblotting

For generating lysates suitable for Western blot, ESCs were cultivated under various conditions (e. g., differentiation) in 6-well, 12-well, or 24-well gelatin-coated plates. After various conditions were met cells were quantified using 0.2% trypan blue to distinguish viable from nonviable cells. Then, up to $4 \times 10^6$ cells were directly lysed via addition of 2x Laemmli Sample Buffer (Bio-Rad) supplemented with 5% β-mercaptoethanol (Millipore Sigma). Lysates were heated at 95°C for five mins, then cooled to room temperature and routinely stored at −20°C. Lysates were loaded into gels such that either the absolute number of viable cells (quantified by trypan blue) or amount of protein (quantified by the Pierce BCA Assay, Thermo Scientific) loaded in each well was the same. Up to 15 µL of lysate was run on a 4–20% Mini-PROTEAN TGX Stain-Free protein gel (Bio-Rad) or a 10% TGX FastCast gel (Bio-Rad) in denaturing conditions at 130V for 50–70 mins followed by semi-dry transfer using the Trans-Blot Turbo Transfer System (Bio-Rad) onto 0.2 µm nitrocellulose or methanol-activated PVDF membranes (Bio-Rad). Successful protein transfer was verified with Ponceau S staining (Amresco) or stain-free fluorescent crosslinking.

After blocking with 5% bovine serum albumin (BSA) or 5% skim milk (for Yap1 and β-actin only) in Tris-buffered saline containing 0.1% Tween 20 (TBST), membranes were incubated overnight with primary antibodies, diluted in 5% BSA as described in the paragraph below. The following day, membranes were washed, incubated with secondary antibodies, washed again, incubated with Amersham ECL Prime Western Blotting Detection Reagent (GE Healthcare) and visualized on a ChemiDoc XRS+ (Bio-Rad). β-actin was used as a loading control.

Primary antibodies (purchased from Cell Signaling Technology unless otherwise specified) along with dilutions used were the following: β-actin (Abgent #AM1829b, 1:20,000), Yap1 (Santa Cruz Biotechnology #sc-101199, 1:1000), Casp8 (#4927S, 1:1000), Casp3 (#9662S, 1:1000), Cleaved Caspase-

3 (#9661S, 1:1000), Cleaved Parp1 (#9548S, 1:1000), Caspase-9 (#9508S, 1:1000), Bcl-2 (#3498S, 1:1000), Bcl-xL (#2764S, 1:1000), Mcl-1 (#94296S, 1:1000), Tead4 (Abcam #ab58310, 1:5000), Puma (Santa Cruz Biotechnology #sc-374223, 1:500), and Bmf (Bioss #bs-7587R, 1:1000). HRP-conjugated secondary antibodies (purchased from Cell Signaling Technology), used at a dilution of 1:10,000 in TBST, were horse anti-mouse (#7076P2) and goat anti-rabbit (#7074S).

## Lentiviral production and infection and transposon-mediated gene integration

Lentiviruses were used to transduce shRNA and overexpression constructs. Bacterial glycerol stocks containing the appropriate shRNA were purchased from Millipore Sigma. A complete list of shRNA can be found in supplemental table S3. HEK293T cells were seeded at a density of $1.2 \times 10^6$ cells per well on a six well plate. After reaching a confluency of 50–60%, cells were transfected with 1.2 µg of an shRNA-containing pLKO-puro vector (Millipore Sigma) as well as 800 ng pCMV-Δ8.9 and 400 ng VSVG packaging plasmids with Fugene 6 (Promega) using the manufacturer's protocol. For inducible overexpression (OE), a pLVX-IRES-ZsGreen1 vector (Clontech) containing the gene of interest and pLVX-TRE3G vector (Clontech) were transfected separately with packaging plasmids. After 18 hr of overnight incubation, HEK293T medium was replaced with ES medium. Then, two days after transfection, medium was supplemented with HEPES to a final concentration of 15 mM to act as an additional buffer, and the supernatant (which contains lentiviral particles) was filtered through a 0.45-µm Supor membrane (PALL). ESCs were infected at a density of $2 \times 10^5$ cells/mL in medium supplemented with 10 µg/mL polybrene (Millipore). 48 hr post-infection, ESCs were selected with puromycin (Thermo Scientific) or geneticin/G418 (Thermo Scientific). Given that cells were passaged every two days, relevant experiments were performed within five passages of the initial infection.

As an alternative method for inducible OE, for the *Wwtr1* gene (Taz), a pSBtet-GP vector (AddGene) with luciferase replaced by a multiple cloning site and cloned with the gene of interest. The resultant construct was transfected along with a transposase-containing pCMV(CAT)T7-SB100 vector (AddGene) into ESCs at a density of $6 \times 10^5$ cells/mL. Selection with puromycin occurred 24 hr later. All relevant experiments were performed within five passages of the initial transfection. Doxycycline (Fisher Scientific) was used at a concentration of 500 ng/mL for all inducible OE experiments. cDNAs for all OE experiments were obtained from either vectors (Bcl-xL - Sino Biological, Bcl-2–3149 pSFFV-neo Bcl-2 cDNA from AddGene, Puma - GenScript, Taz - pcDNA3.1/HisC-mTAZ from AddGene) or full-length mouse ESC cDNA reverse transcribed using the ProtoScript II First Strand cDNA Synthesis Kit from New England Biolabs (Bmf). All inserts were confirmed by Sanger sequencing.

## Caspase activity assay

For determination of caspase activity, the Caspase-Glo 3/7, 8, and 9 Assay Systems (Promega) was used. ESCs were seeded at a density of $1 \times 10^5$ cells/mL in a white-walled 96-well plate (Millipore Sigma). At the indicated timepoints, cells were assayed using the respective kits according to manufacturer's instructions. After subtracting the noise from blank wells (containing media but no cells), luminescent signals in each well were normalized to the cell number.

## Gene expression analysis

RNA-seq data was downloaded from Gene Expression Omnibus. Yap1 KD data belonged to the series GSE69669. Samples corresponding to accession numbers GSM1706496, GSM1706495, GSM1706489, and GSM1706488 were used for analysis. Boxplots were generated using BoxPlotR (http://shiny.chemgrid.org/boxplotr/) where whiskers extend to the 5th and 95th percentiles. Gene lists were taken from AmiGO 2 (http://amigo.geneontology.org), specifically positive (GO:2001244) and negative (GO:2001243) regulation of intrinsic apoptotic signaling pathway. Lists were double-checked for any genes known to behave differently than annotated in the ES cell context. Genes that were not expressed were removed from the analysis to reduce noise. Gene ontology analysis was carried out using GOrilla (http://cbl-gorilla.cs.technion.ac.il/) using the 'two unranked lists of genes' option, where the background list was populated by all genes listed in the RNA-seq output.

For RT-qPCR, total RNA was extracted from cells with the RNeasy Plus Mini Kit (Qiagen). Then, 600 ng of RNA was reverse transcribed into cDNA using the qScript cDNA SuperMix from Quanta-Bio (VWR). Next, qPCR was performed in 20 µL reactions using the PerfeCTa SYBR Green FastMix

(VWR) plus 6 ng of cDNA and 250 nM forward and reverse primers. Primers were designed using Primer3 (http://primer3.ut.ee/) such that each primer amplified the junction between two or more exons, and their specificity as well as lack of primer dimer formation was verified with melt curve analysis showing one peak. Relative expression was normalized to Gapdh using the $2^{-\Delta\Delta CT}$ method. All reactions were performed at least in triplicate on a StepOnePlus Real-Time PCR System (Applied Biosystems). All primer sequences are listed in Supplemental Table S1.

## Immunofluorescence

ESCs were seeded ($6 \times 10^5$ cells/mL) on a gelatin-coated μ-Slide VI 0.4 (Ibidi). For cells growing in -LIF conditions, cells were differentiated on a 10 cm plate for one day before seeding the μ-Slide and seeded at $2 \times 10^5$ cells/mL. After an additional 2 days, cells were thoroughly washed with Dulbecco's phosphate-buffered saline (DPBS) and fixed using 4% paraformaldehyde (freshly cracked with 70 mM NaOH at 70°C) for 15 mins. For mitochondrial staining, after washing but before fixation, cells were incubated in 300 nM MitoTracker Deep Red FM in OptiMem for 30 mins. Cells were washed again with DPBS and then permeabilized with 0.3% Triton X-100 in PBS for five mins. After washing once more, samples were blocked using IF blocking solution (4% BSA and 1% normal goat serum diluted in DPBS) for one hour, then incubated overnight with Bcl-2 (1:200) or Mcl1 (1:800) primary antibodies diluted in IF blocking solution. Then, samples were washed thoroughly followed by incubation with fluorescent secondary antibody (goat anti-rabbit IgG Alexa Fluor 594 from Thermo Scientific) diluted 1:1000 for one hour. After further washing, ProLong Glass Antifade Mountant with NucBlue (Thermo Scientific) was added to the samples, which were allowed to cure for 18–24 hr at room temperature. Slides were then imaged using a Zeiss LSM 710 Confocal Microscope using the Plan-Apo 63X (oil) objective and images were processed using ZEN microscope software. Colocalization was quantified using Zen software by setting the crosshairs such that noise was restricted to the lower left quadrant, and the same crosshair coordinates were used for all samples. Intensity was quantified using ImageJ and normalized to the number of discrete nuclei (stained by NucBlue) that could reasonably be assigned to separate cells.

## Chromatin immunoprecipitation followed by NextGen sequencing (ChIP-seq)

After reaching ~80% confluency in a 15 cm plate, BirA ESCs with or without FLAG-Bio-Yap1 were crosslinked with 1% formaldehyde for 7 min at room temperature and constant shaking. Formaldehyde was quenched with addition of glycine to a final concentration of 125 mM along with shaking for 5 min. Cells were then sonicated using a Bioruptor (Diagenode), and sheared chromatin including DNA fragments ~ 300 bp in length were used for immunoprecipitation with Dynabeads MyOne Streptavidin T1 (Thermo Scientific). Sequencing libraries were prepared with the enriched ChIP sample using the NEBNext Ultra II DNA Library Prep Kit for Illumina (New England Biolabs) and sequenced using the Illumina HiSeq 4000 at the UT Austin Genomic Sequencing and Analysis Facility (GSAF).

## ChIP-seq data analysis

Public ChIP-seq data sets were downloaded from Cistrome (http://cistrome.org/db/#/). When possible, only data sets that passed all of Cistrome's quality control conditions were used. To determine pairwise correlations between ChIP-seq data sets, human YAP1 ChIP-seq peaks.bed files were sent directly to Galaxy (http://cistrome.org/ap/) and peaks were assigned to genes using the BETA-minus functionality (assembly hg19). For mouse Tead factor ChIP-seq datasets and our own Yap1 ChIP-seq data, fastq files were directly processed using the SRA Toolkit, and 75 bp reads were mapped onto the mouse genome (assembly mm9) using Bowtie 2. Peaks were then called using model-based analysis of ChIP-seq (MACS2). For comparison with p300 ChIP-seq data, Bowtie two output was used to compare target overlap within a window of 6 kb of the Yap1 peak center (in dESCs) using a bin size of 100 bp. The apoptosis gene list was retrieved from AmiGO 2 (GO:0006915). Binding scores for all genes were then used for pairwise correlations using Vassar Stats (http://vassarstats.net/matrix2.html) and correlations were visualized using Java TreeView (http://jtreeview.sourceforge.net/). Signal tracks were visualized using Integrated Genome Viewer (IGV, http://software.broadinstitute.org/software/igv/).

For our own Yap1 ChIP-seq data, motif analysis was performed using HOMER (http://homer.ucsd.edu/homer/motif/fasta.html). Peak to gene features were assigned using in-house Perl code. Binding sites were assigned to genomic features according to the following hierarchy: promoter (±2 kb of the TSS)>upstream (2–20 kb upstream of the TSS)>intron > exon>intergenic (all other binding sites that did not fit the other categories). Gene ontology (GO) analysis was performed using GOrilla (http://cbl-gorilla.cs.technion.ac.il/) using the two unranked lists of genes (target and background lists) setting. For the target list, all the genes with a peak score (normalized to BirA) greater than two were included. These Yap1 target genes were further sorted into either upregulated upon Yap1 KD (log2(KD/control)≥0.5) or downregulated ((log2(KD/control)≤−0.5). For the background list, all the genes from the bed file (20,422) were included. The top five GO terms (relative to -log10(p-value)), plus the top apoptosis-related GO term, were then graphed.

## Dual luciferase reporter assay

ESCs were seeded ($6 \times 10^5$ cells/mL when comparing Yap1 OE to empty, $4 \times 10^5$ cells/mL when comparing Yap1 KO to WT) in a white-walled 96-well plate. Cells were transfected with 40 ng pGL3-promoter (Promega) containing firefly luciferase downstream of the SV40 promoter plus putative Yap1-responsive regulatory elements cloned from genomic mouse DNA. Simultaneously, as an internal control, cells were co-transfected with 40 ng pRL-TK containing Renilla luciferase downstream of the HSV-thymidine kinase promoter. During Yap1 OE experiments, half of the wells were transfected with 40 ng of a FLAG-Bio vector containing either Yap1, mutant Yap1 (Ser79Ala) generated by site-directed mutagenesis via the NEBuilder HiFi DNA Assembly Kit (New England Biolabs), or no insert downstream of the EF-1α promoter. All transfections related to luciferase were performed with Lipofectamine 3000 (Life Technologies). Cells were incubated for 16 hr before changing the media, and luciferase activity was measured by the Dual-Glo Luciferase Assay a total of 24 hr after transfection System (Promega). Firefly luciferase signal was normalized to Renilla luciferase signal, and then the signal of each regulatory element-containing construct was normalized to pGL3-promoter. All regulatory element sequences tested are listed in Supplemental Table S2. The NEBuilder HiFi DNA Assembly Kit was used to assemble the Bcl-2 tandem enhancer as well as the Mcl-1 distal enhancer with Tead site deletion (using overlapping homology excluding the Tead binding motif).

## Mitochondrial priming and loss of membrane potential

Mitochondrial membrane potential loss (Δψ) was measured as a change in the 525/570 nm ratio relative to the DMSO-treated control using the Cell Meter JC-10 Mitochondrion Membrane Potential Assay Kit (AAT Bioquest) according to the manufacturer's instructions after 12 hr of incubation with either BH3 mimetic or various timepoints of differentiation (72 hr for -LIF and EpiLC, 48 hr for neural ectoderm and endoderm). BH3 mimetics ABT-737 (*Oltersdorf et al., 2005*), Venetoclax/ABT-199 (*Souers et al., 2013*), A-1210477 (*Leverson et al., 2015*), and A-1155463 (*Tao et al., 2014*) were applied to ESCs or dESCs (after 24 hr of differentiation) at the concentrations indicated in the figure. Cell death was measured using the LDH assay as described above after 24 hr of incubation with the BH3 mimetic (48 hr total after LIF withdrawal).

## siRNA knockdown

MISSION siRNA was purchased from Millipore Sigma. Duplexes targeting *Mcl1* (NM_008562: SASI_Mm01_00048593, SASI_Mm02_00314161, SASI_Mm01_00048594) as well as *Bcl2l1* (NM_009743: SASI_Mm02_00316924, SASI_Mm02_00316925, SASI_Mm02_00316926) were ordered and resuspended at a concentration of 25 µM in 5X siRNA buffer (Dharmacon) diluted to 1X RNase-free water (Thermo Scientific). siRNA was reverse transfected into ESCs ($6 \times 10^5$ cells/mL) at a final concentration of 75 nM using INTERFERin according to the manufacturer's protocol (Polyplus Transfection). MISSION siRNA Fluorescent Universal Negative Control #1 conjugated to 6-FAM was used as both a transfection control and as a non-targeting siRNA control. After verifying KD at the protein level by Western blot, the best two siRNAs were chosen for further experiments. All shRNA and siRNA TRC/ID numbers, and shRNA sequences (or the target position where siRNA is predicted to bind) are listed in supplemental table S3.

## Data, software, and code availability

Yap1 ChIP-seq data generated in this study has been uploaded to Gene Expression Omnibus under accession number GSE112606. Code used to analyze raw sequencing files using the programs STAR, Bowtie2, MACS, and Homer is available in the code file included with this manuscript (Source code file 1).

## Acknowledgements

This study was supported by R01GM112722 (NIH) and the Preterm Birth Research Grant (Burroughs Welcome Fund) to JK, as well as the NSF GRFP and Hamilton Seed Grant to LL. We thank the Genome Sequencing and Analysis Facility (GSAF) and Texas Advanced Computing Center (TACC) at UT Austin for ChIP-seq analysis as well as the Center for Biomedical Research Support at UT Austin for flow cytometry and confocal microscopy.

## Additional information

### Funding

| Funder | Grant reference number | Author |
|---|---|---|
| National Institute of General Medical Sciences | R01GM112722 | Jonghwan Kim |
| Burroughs Wellcome Fund | | Jonghwan Kim |
| National Science Foundation | GRFP | Lucy LeBlanc |
| Hamilton Seed Grant | Departmental Grant from Molecular Biosciences | Lucy LeBlanc |

The funders had no role in study design, data collection and interpretation, or the decision to submit the work for publication.

### Author contributions

Lucy LeBlanc, Conceptualization, Resources, Formal analysis, Funding acquisition, Validation, Investigation, Writing—original draft, Writing—review and editing; Bum-Kyu Lee, Formal analysis, Investigation; Andy C Yu, Mijeong Kim, Aparna V Kambhampati, Shannon M Dupont, Davide Seruggia, Byoung U Ryu, Investigation; Stuart H Orkin, Supervision, Investigation; Jonghwan Kim, Conceptualization, Resources, Data curation, Formal analysis, Supervision, Funding acquisition, Investigation, Writing—original draft, Project administration, Writing—review and editing

### Author ORCIDs

Lucy LeBlanc http://orcid.org/0000-0001-5945-3133
Jonghwan Kim http://orcid.org/0000-0002-9919-9843

### Decision letter and Author response

Decision letter https://doi.org/10.7554/eLife.40167.034
Author response https://doi.org/10.7554/eLife.40167.035

## Additional files

### Supplementary files

• Source code 1. Code used to analyze raw sequencing files using the programs STAR, Bowtie2, MACS, and Homer.
DOI: https://doi.org/10.7554/eLife.40167.016

• Supplementary file 1. Supplementary Table S1. Table of RT-qPCR primers used for qPCR gene expression assays in this study. Primers were designed using Primer3 and verified by melt curve analysis. Supplementary Table S2. Table of cloning primers used for dual luciferase assay including

chromosome coordinates (using mm9) and regulatory element length. Supplementary Table S3. Table of shRNA and siRNA used in KD experiments including target, ID, and sequence or target position.

DOI: https://doi.org/10.7554/eLife.40167.017

• Transparent reporting form

DOI: https://doi.org/10.7554/eLife.40167.018

## Data availability

Sequencing data have been deposited in GEO under accession code GSE112606.

The following dataset was generated:

| Author(s) | Year | Dataset title | Dataset URL | Database and Identifier |
|---|---|---|---|---|
| Bum-Kyu Lee, Lucy LeBlanc, Jonghwan Kim | 2018 | Yap1 safeguards mouse embryonic stem cells from excessive apoptosis during differentiation | https://www.ncbi.nlm.nih.gov/geo/query/acc.cgi?acc=GSE112606 | NCBI Gene Expression Omnibus, GSE112606 |

The following previously published datasets were used:

| Author(s) | Year | Dataset title | Dataset URL | Database and Identifier |
|---|---|---|---|---|
| Diepenbruck M, Waldmeier L, Ivanek R, Berninger P, Arnold P, van Nimwegen E, Christofori G | 2014 | Tead2 expression levels control the subcellular distribution of Yap and Taz, zyxin expression and epithelial-mesenchymal transition. | https://www.ncbi.nlm.nih.gov/geo/query/acc.cgi?acc=GSE55709 | NCBI Gene Expression Omnibus, GSE55709 |
| Zanconato F, Forcato M, Battilana G, Azzolin L, Quaranta E, Bodega B, Rosato A, Bicciato S, Cordenonsi M, Piccolo S | 2015 | Genome-wide association between YAP/TAZ/TEAD and AP-1 at enhancers drives oncogenic growth. | https://www.ncbi.nlm.nih.gov/geo/query/acc.cgi?acc=GSE66081 | NCBI Gene Expression Omnibus, GSE66081 |
| Stein C, Bardet AF, Roma G, Bergling S, Clay I, Ruchti A, Agarinis C, Schmelzle T, Bouwmeester T, Schübeler D, Bauer A | 2015 | YAP1 Exerts Its Transcriptional Control via TEAD-Mediated Activation of Enhancers. | https://www.ncbi.nlm.nih.gov/geo/query/acc.cgi?acc=GSE61852 | NCBI Gene Expression Omnibus, GSE61852 |
| Chung H, Lee BK, Uprety N, Shen W, Lee J, Kim J | 2016 | Yap1 is dispensable for self-renewal but required for proper differentiation of mouse embryonic stem (ES) cells | https://www.ncbi.nlm.nih.gov/geo/query/acc.cgi?acc=GSE69669 | NCBI Gene Expression Omnibus, GSE69669 |
| Hu Y, Zhang Z, Kashiwagi M, Yoshida T, Joshi I, Jena N, Somasundaram R, Emmanuel AO, Sigvardsson M, Fitamant J, El-Bardeesy N, Gounari F, Van Etten RA, Georgopoulos K | 2016 | Superenhancer reprogramming drives a B-cell-epithelial transition and high-risk leukemia. | https://www.ncbi.nlm.nih.gov/geo/query/acc.cgi?acc=GSE86897 | NCBI Gene Expression Omnibus, GSE86897 |
| Obier N, Cauchy P, Assi SA, Gilmour J, Lie-A-Ling M, Lichtinger M, Hoogenkamp M, Noailles L, Cockerill PN, Lacaud G, Kouskoff V, Bonifer C | 2016 | Cooperative binding of AP-1 and TEAD4 modulates the balance between vascular smooth muscle and hemogenic cell fate. | https://www.ncbi.nlm.nih.gov/geo/query/acc.cgi?acc=GSE79320 | NCBI Gene Expression Omnibus, GSE79320 |

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
