## [Decision Letter]

Thank you for sending your article entitled "Yap1 safeguards mouse embryonic stem cells from excessive apoptosis during differentiation" for peer review at *eLife*. Your article is being evaluated by three peer reviewers, and the evaluation is being overseen by a Reviewing Editor and Marianne Bronner as the Senior Editor.

As you will see, the reviewers think the paper is potentially very interesting and well-executed but also feel like some major revisions would be required. There was concern by some reviewers that in vivo validation would take too long and may not be necessary, so we would appreciate hearing your opinion in that regard. We hope you find the reviews helpful.

*Reviewer #1:*

In this manuscript the authors analyzed the role of YAP1 during differentiation of stem cells in culture. For this purpose, endogenous YAP1 was deleted using Crispr/Cas technology. The authors demonstrate that while 30% of the control WT cells die after exiting self-renewal, up to 70% of YAP1 KO cells die via apoptotic cell death. YAP1 was necessary for prominent upregulation of anti-apoptotic genes and attenuation of pro-apoptotic genes during stem cell differentiation and this plays an important role for protection of differentiating stem cells from apoptotic cell death. Authors also demonstrate that forced overexpression of anti-apoptotic BCl^-^xl or Bcl2 in YAP1 KO cells rescue cell death upon differentiation. The authors propose that YAP1 directly regulates pro- and anti-apoptotic genes to protect stem cells from apoptotic cell death during differentiation.

Extensive literature implicates YAP1 in protection of cells from apoptotic cell death. This manuscript extends this knowledge and shows that YAP1 is necessary to protect cultured stem cells from apoptotic cell death during differentiation. While this finding is well documented, the molecular mechanisms responsible for this YAP1 function are not completely clear. The physiological relevance of this mechanism in a live organism is also not obvious.

Specific points:

1) Physiological relevance. Differentiation induced by growth factor withdrawal in culture is a somewhat artificial process. It is not clear if such an extensive activation of apoptosis happens during differentiation of stem cells in vivo and whether YAP1 protects from it. To add physiological relevance, authors should have analyzed YAP1-/- embryos.

2) Mechanism. It still not clear if YAP1 is playing an instructive or permissive role in protection from apoptosis. The authors show tremendous upregulation of Bcl2, BCl^-^xl and MCl^-^1 during differentiation of control wildtype stem cells and the lack of this upregulation in YAP1 KO cells (Figure 2A). Overexpression of Bcl2 or BCl^-^xl rescues YAP1 KO cells (Figure 4DE). YAP1 is binding to Bcl2 and other anti-apoptotic genes and the YAP1-binding elements from these genes better activate Luciferase in wild-type than in YAP1 KO cells. It appears that YAP1 directly activates anti-apoptotic genes which protect cells from apoptotic death. On the other hand, the authors show very little upregulation of Bcl2 and BCl^-^xl in cells overexpressing YAP1 (Figure 3E); however, the apoptosis phenotype is rescued completely by exogenous YAP1(Figure 4D). If YAP1 activation directly upregulates Bcl2 and BCl^-^xl to protect cells from apoptosis, why there is a complete rescue without significant changes in Bcl2 and BCl^-^xl expression?

3) Figure 2. Are all these changes in YAP1 KO cells the drivers or the consequences of apoptotic process? What happens is apoptosis is chemically induced in normal WT ES cells and then the ratios of anti- and pro-apoptotic mRNAs is compared between apoptotic and non-apoptotic cells? If the same changes are observed in that experiment, then these are likely the consequences of apoptosis, not the drivers.

4) Subsection “Yap1 modulates the expression of apoptosis-related genes during differentiation”; "Meanwhile, constitutive Yap1 OE slightly induced anti-apoptosis genes on average (Figure 2—figure supplement 2C). These data suggest that Yap1 may be a master regulator of anti-apoptotic genes during ESC differentiation." Is the difference significant in Figure 2—figure supplement 2C? If it is not significant, then the statement is not completely correct.

5) In the same section; "Intriguingly, analyzing publicly available RNA-seq data from four human cell types using gene set expression analysis (GSEA) showed dysregulation of genes involved in apoptotic signaling after YAP1 KD relative to control KD, showing that Yap1 may transcriptionally regulate apoptosis in both mouse and human cells (Figure 2—figure supplement 2D). Is the GSEA overlaps significant in Figure 2—figure supplement 2D? What are the p and q values? If it is not significant, then the statement is not completely correct.

6) Subsection “Yap1 directly regulates apoptosis-related genes via transcription”: "Meanwhile, transient OE of Yap1 led to higher luciferase activity with anti-apoptotic gene regulatory elements and lower luciferase activity with pro-apoptotic gene regulatory elements (Figure 3E)." What happens to endogenous Bcl2 and BCl^-^xl genes in YAP1 overexpressing cells? They should be prominently upregulated, if the authors model is correct?

7) Figure 3D. Is TEAD-YAP1 -binding necessary for upregulation of Luciferase expression in Figure 3D? This can be analyzed by mutating TEAD-YAP-binding consensus sequences in these constructs. If YAP1 role is direct, TEAD-YAP binding to the promoter should be critical.

8) Figure 2 is not well organized. What happens to pro-apoptotic transcripts during differentiation in WT ES cells? Are they up or downregulated in comparison to non-differentiated cells? It appears that the direction of the changes in gene expression are similar in WT and YAP KO cells, but only the magnitude is different. This is not obvious the way the data are presented.

9) Figure 3C. Is there any change in YAP1 binding to pro- and anti-apoptotic genes during differentiation? This static data are not very informative. Especially considering that some YAP1 binding genes are upregulated while others are downregulated.

10) Figure 4C should have data for Bmf and Puma knockdowns in WT cells. If the difference between WT and YAP1 KO cells significantly reduced upon Bmf and Puma knockdowns in both cell types, then Bmf and Puma genes play a specific role. If the difference between WT and KO cells is not significantly reduced upon Bmf and Puma knockdowns in both cell types, then Bmf and Puma genes do not play a specific role.

11) Figure 1—figure supplement 1J. Knockdown of what gene?

*Reviewer #2:*

This is an interesting study that tries to address the poorly explored phenomena that about 30% of ESCs undergo apoptosis upon differentiation initiation. The authors show convincingly that YAP1 keeps these levels low relatively, and KO of Yap1 increases apoptosis through increase caspase mediate cell death and mitochondrial priming. They show that YAP1 directly regulates transcription of key anti apoptotic genes.

Overall this is an interesting study, however some important issues need to be examined and clarified before publication:

1) Please show rescue of phenotype in YAP1 KO ESCs upon ectopic expression of Yap1 in order to exclude off target phenotypes.

2) Does overexpression of YAP1 in WT ESCs further decreases apoptosis upon LIF removal ?

3) What about the other hippo pathway effector – can TAZ overexpression rescue the phenotype of YAP1 KO? Does TAZ overexpression in WT ESCS reduce apoptosis upon LIF withdrawal?

4) The authors indicate "Thus, Yap1 is key for survival during ESC differentiation, and ablation of Yap1 exacerbates apoptosis specifically rather than triggering an alternate cell death pathway." However, they do not show survival rate of yap KO in a few passages in differentiated state. How do they know lethality is during the differentiated state transition, and not just more lethal in the differentiated state itself?

5) The authors indicate "This implied that the abnormally high rates of apoptosis in Yap1 KO cells are sustained by heightened Casp9 activation. However, since mRNA expression of caspases was relatively equal between Yap1 KO cells and WT cells during differentiation, we speculated that Yap1 may regulate other factors that indirectly affect the rate of caspase cleavage". This is over speculation. If mRNA levels are the same, protein levels should be tested.

6) The authors indicate "Immunocytochemistry shows that Yap1 nuclear localization tends to be associated with higher expression of BCl^-^2 in -LIF, but not in +LIF conditions, where undifferentiated ESCs show cytoplasmic Yap1 and weaker BCl^-^2 staining" (Figure 2—figure supplement 2A)". The ICC looks of poor quality and looks like a multicellular effect and not a single cell phenotype. Can this be explained or improved?

7) The authors indicate "Collectively, these data show that Yap1 is critical for proper BCl^-^2 induction during differentiation". However Yet in Figure 3E they show no change in BCL2 luci with OE of YAP? Can this be better explained and addressed.

8) The authors indicated "Using RNA-seq data from a previous study (Chung et al., 2016), we found that differentiation induces the expression of a group of anti-apoptotic genes in WT cells, but this induction is debilitated after Yap1 KD (Figure 2—figure supplement 2B)." No t-test performed for any of these boxplots in Figure 2—figure supplement 2. This should be added and verified.

9) The authors indicate "Additionally, we confirmed a physical interaction between Yap1 and Tead4 during ESC differentiation using co-immunoprecipitation (Figure 3—figure supplement 3D)." Did the authors conduct an IP for p300 as well, if they show a gene correlation in ChIP? This is also what they present in the end Figure (5), but they don't show any IP for it to substantiate this claim?

10) The authors indicate "Furthermore, deletion of Yap1 significantly sensitized ESCs, but not undifferentiated ESCs, to loss of Δѱ in response to BH3 mimetic treatment. We then investigated whether the higher loss of Δѱ in Yap1 KO cells correlated with greater rates of cell death". I wonder why did the authors use only response to BH3 treatment? Why not +/- LIF like before? Could this be also shown and explained.

*Reviewer #3:*

Yap1 plays numerous roles in development and cancer but its roles during ESC differentiation and early development remains unclear. Previous studies showed the death of 30% or more of ESCs upon differentiation and proposed that apoptosis during ESC differentiation is to cull cells that fail to exit self-renewal in order to promote efficient differentiation. This function of apoptosis is not limited to ESC differentiation, as it also happens in other biological processes. However, mechanisms that regulate the balance between survival and death during ESC differentiation remain unclear. In this manuscript, by using the Yap1 KO ESCs established in their previous studies, LeBlanc et al. show that Yap1 KO ESCs experience massive cell death upon the exit from self-renewal and Yap1 contextually protects differentiating, but not self-renewing ESC from hyperactivation of the apoptotic cascade. Yap1 attenuates mitochondrial apoptosis during ESC differentiation by upregulating anti-apoptotic factors through transcriptional regulation and Yap1 KO cells develop a high degree of mitochondria priming that precedes elevated rates of apoptosis.

This study is well executed and timely, and the message is clear and significant in contributing to our understanding of the functions of Yap1 during ESC differentiation and early development and of the balance control between survival and death during cell fate changes. While I am strongly supportive for its publication in *eLife*, I do have some major and minor points listed below that could further enhance the clarity and improve the manuscript. And the authors should pay more attention to the labeling in the figures and provide a few minimal additional experiments and more explanations to the experiment design to make the overall story more straightforward and clearer.

Major points:

1) At beginning the authors tested several mESC differentiation models (LIF withdraw, N2B27, and IDE1). However, it will be nicer if they also test cell apoptosis of WT/Yap1KO ESCs in EpiLC differentiation, because 1) EpiLC differentiation mimics the in vivo epiblast maturation at pre-to-post implantation stage, and 2) it is more related to the context of Yap1 and p300 ChIP-seq analysis as performed in Figure 3.

2) Figure 1 shows the upregulation of Casp and apoptosis in Yap1 KO starts early (24-48 hrs) after LIF withdrawal, but Figure 2A and 2B show the upregulation of anti-apoptotic factors (BCl^-^2, BCl^-^xL and MCl^-^1) in WT ES starts late (72 hrs) after LIF withdrawal. BCl^-^2 is the regulator upstream of Casp. The authors should provide some explanations about these results.

3) In the Figure 1E, 1F and 2D, please indicate the time after LIF withdrawal (or the time of induced differentiation) in hours to better compare them with other results (like Figure 1D, 2A, 2B). I didn't find their time point information in Figure/Figure legends/Text/Experimental procedures.

4) About Figure 3D and 3E: (1) the authors should at least provide one representation of the reporter construct; (2) the authors should indicate the regulatory elements (e.g., by using bars) used in the luciferase constructs in the Yap1 and p300 ChIP-seq tracks; (3) I am curious to see the expression level of Yap1 in Yap1 OE: was it rescued to a WT level or an overexpression?

5) Figure 3 – data are clear but too shallow/superficial: According to the increased p300 (and H3K27ac) ChIP-seq intensities at Yap1 peaks during differentiation, it is implied that Yap1 may function as an activator of transcription for the anti-apoptotic genes BCl^-^2, BCl^-^xL, MCl^-^1. ChIP-qPCR are required to confirm the Yap1 binding at those loci (Figure 3C). Furthermore, authors should also compare the Yap1 and p300 ChIP intensity between ES and dES situations, to support their conclusion that Yap1 activates the anti-apoptotic genes during differentiation. Does Yap1 physically interact with p300? If so, the authors may want to show if Yap1 recruits p300 to chromatin.

6) Figure 4C: the label is misleading. In the figure, the authors should indicate the Bmf KD and Puma KD are performed in Yap1 KO.

---

## [Author Response]

We appreciate all the reviewers for their insightful and constructive comments. We recognize that the top concern is a lack of clarity in elucidating Yap1’s molecular mechanisms in promoting survival via transcriptional regulation. We have attempted to address this concern and others by performing a considerable number of additional experiments to support our claims. The following is a summary of our responses to major or recurring comments:

To address whether Yap1 truly binds more to target apoptosis-related gene loci during differentiation rather than self-renewal, we use both ChIP-seq and ChIP-qPCR to show that Yap1 occupancy indeed increases on target loci (most notably MCl^-^1 and BCl^-^2) after the exit from self-renewal. Additionally, to further characterize enhancers linked to apoptosis-related genes, we generated a tandem BCl^-^2 enhancer that is more responsive to OE of Yap1 than our previous enhancer, deleted the Tead-binding consensus sequence from the MCl^-^1 enhancer, and demonstrated that OE of a mutant Yap1 that cannot bind to Tead factors is deficient in rescue of luciferase expression. Finally, in response to concerns that in vivovalidation was lacking, we performed several experiments using the EpiLC differentiation condition in vitro, which more closely mimics the transition from inner cell mass to epiblast that occurs in vivo, compared to mere LIF withdrawal.

Reviewer #1:

[…] Extensive literature implicates YAP1 in protection of cells from apoptotic cell death. This manuscript extends this knowledge and shows that YAP1 is necessary to protect cultured stem cells from apoptotic cell death during differentiation. While this finding is well documented, the molecular mechanisms responsible for this YAP1 function are not completely clear. The physiological relevance of this mechanism in a live organism is also not obvious.

We sincerely appreciate the reviewer’s constructive feedback for the improvement of our manuscript. We have performed new experiments that hopefully address the reviewer’s major concerns and illuminate the molecular mechanisms involved.

Specific points:

*1) Physiological relevance. Differentiation induced by growth factor withdrawal in culture is a somewhat artificial process. It is not clear if such an extensive activation of apoptosis happens during differentiation of stem cells* in vivo *and whether YAP1 protects from it. To add physiological relevance, authors should have analyzed YAP1-/- embryos.*

We thank you for this critical comment. Yap1 -/- embryos have been made previously and show morphological defects by E7.5 and developmental arrest by E8.5, culminating in embryonic lethality by E10.5 (Morin-Kensicki et al., 2006). Key defects included yolk sac vasculogenesis and failure of the allantois to attach to the chorion, which implicated Yap1 in regulating cell number, morphogenetic movements, or cell-cell interactions, but the authors could not pin down any concrete mechanism as to why Yap1 -/- led to such broad developmental defects. Thus, we agree that generating and analyzing our own Yap1 -/- embryos in greater mechanistic depth would have been a considerable contribution.

However, we bring up the comment from reviewer 3, who helpfully suggested epiblast-like cell (EpiLC) differentiation as a way to more closely reflect what occurs in vivo. The ESC to EpiLC cell fate conversion is reminiscent of the in vivo transition of the inner cell mass (ICM) to the epiblast, and in vitro differentiation techniques produce EpiLCs that are highly similar to epiblasts according to principal component analysis of RNA-seq data (Hayashi et al., 2011). We performed this differentiation method and analyzed cell death, apoptosis-related gene expression, and mitochondrial depolarization, finding them to be relatively consistent with differentiation experiments utilizing LIF withdrawal (new results presented in Figures 2D, 3B, and 5A).

Furthermore, stem cell therapy often requires culturing ESC or patient-specific iPS cells and differentiation of them in vitro, before transplantation to treat various diseases (Jiaojiao et al., 2018; Shi et al., 2016). Therefore, we believe thatin vitrostudies of ESC differentiation are still valuable because of their application to biotechnological and biomedical advances.

2) Mechanism. It still not clear if YAP1 is playing an instructive or permissive role in protection from apoptosis. The authors show tremendous upregulation of Bcl2, BCl^-^xl and MCl^-^1 during differentiation of control wildtype stem cells and the lack of this upregulation in YAP1 KO cells (Figure 2A). Overexpression of Bcl2 or BCl^-^xl rescues YAP1 KO cells (Figure 4DE). YAP1 is binding to Bcl2 and other anti-apoptotic genes and the YAP1-binding elements from these genes better activate Luciferase in wild-type than in YAP1 KO cells. It appears that YAP1 directly activates anti-apoptotic genes which protect cells from apoptotic death. On the other hand, the authors show very little upregulation of Bcl2 and BCl^-^xl in cells overexpressing YAP1 (Figure 3E); however, the apoptosis phenotype is rescued completely by exogenous YAP1(Figure 4D). If YAP1 activation directly upregulates Bcl2 and BCl^-^xl to protect cells from apoptosis, why there is a complete rescue without significant changes in Bcl2 and BCl^-^xl expression?

This comment makes an excellent point, as the aforementioned data do seem to be somewhat contradictory. First, we should clear up some misconceptions. We showed upregulation of only BCl^-^2 during differentiation; our new Figure 2B shows that BCl^-^xL and MCl^-^1 are either slightly upregulated or merely maintained during differentiation, at least by 48 hr. Yap1 KO cells fail to upregulate BCl^-^2, but they also seem to fail to maintain levels of BCl^-^xL or MCl^-^1.

The seeming lack of upregulation of BCl^-^2 and BCl^-^xL via luciferase expression after OE of Yap1 can be explained by looking at the asymmetric transcriptional effects of KO and OE. Complete lack of Yap1 seems to be more impactful on BCl^-^2 levels than transient overexpression, where a 16-fold induction of Yap1 only upregulated anti-apoptotic genes by approximately 2-fold (new Figure 3—figure supplement 1C). Therefore, there is indeed upregulation of these genes after OE, it was simply not detected by our previous luciferase assays.

Furthermore, enhancers rarely work alone in cells, and Yap1 showed multiple peaks on all target apoptosis-related genes discussed in this study. Thus, we surmised that creating a tandem enhancer consisting of two elements occupied by Yap1 according to ChIP-seq may more closely approximate what is occurring in the cell. Indeed, combining our old BCl^-^2 enhancer with an additional element in the same intron (new Figure 4—figure supplement 1H) granted upon it the ability to increase expression 2x upon Yap1 OE (modified Figure 4E), consistent with our RT-qPCR data without changing basal enhancer activity.

3) Figure 2. Are all these changes in YAP1 KO cells the drivers or the consequences of apoptotic process? What happens is apoptosis is chemically induced in normal WT ES cells and then the ratios of anti- and pro-apoptotic mRNAs is compared between apoptotic and non-apoptotic cells? If the same changes are observed in that experiment, then these are likely the consequences of apoptosis, not the drivers.

We appreciate this comment, as apoptosis-related genes not only contribute to cell death but are in turn regulated by cell death itself. Indeed, it is known that apoptosis initiates global mRNA decay, which contributes to amplifying cell death when anti-apoptotic transcripts are depleted, forming a feedforward loop (Thomas et al., 2015). Complicating this picture, anti-apoptotic proteins can function as transcriptional regulators in and of themselves (Wu et al., 2017). We believe that the best way to disentangle this highly interconnected phenomenon is by modulation of individual genes to genetically dissect requirements for apoptosis in the absence of Yap1.

However, we did perform the requested experiment in an attempt to determine whether changes in ratios of anti- and pro-apoptotic mRNAs are sufficient drive apoptosis. We have chemically induced apoptosis using STS and measured the expression of pro- and anti-apoptotic genes over a timecourse. Surprisingly, we found that genes involved in intrinsic apoptosis (both pro and anti) were strongly repressed over the course of STS-induced apoptosis, whereas *Fas* and *Fasl* which are involved in promoting extrinsic apoptosis were significantly increased (Author response image 1). However, this did not seem to support or refute any of our major claims in our study.

Thus, as an alternative approach to your comment, we proposed that individual repression or overexpression of apoptosis-related genes would determine whether such changes in expression could in and of themselves drive changes in rates of cell death. We had already knocked down or overexpressed most factors, and these single factor manipulations did rescue or exacerbate cell death (Figures 5D, 6A, 6B, 6E, and 6F); to strengthen this point, we inducibly overexpressed pro-apoptotic factors Bmf and Puma. OE accelerated, but did not cause, cell death in dESCs, particularly in Yap1 KO cells, and particularly with Puma, agreeing with our KD data (new Figure 5E, Figure 5—figure supplement 1C and D). We believe that these experiments provide sufficient evidence that at least individually, changes in the expression of these apoptosis-related genes can shift cell fate to either survival or death in dESCs.

**Author response image 1. respfig1:** dESCs were treated with 1 μM STS for the indicated hours and expression of various anti- and pro-apoptotic genes was measured at each treatment timepoint (n = 2). Expression values were normalized to Gapdh and to 0 hr (DMSO-treated) dESCs

4) Subsection “Yap1 modulates the expression of apoptosis-related genes during differentiation”; "Meanwhile, constitutive Yap1 OE slightly induced anti-apoptosis genes on average (Figure 2—figure supplement 2C). These data suggest that Yap1 may be a master regulator of anti-apoptotic genes during ESC differentiation." Is the difference significant in Figure 2—figure supplement 2C? If it is not significant, then the statement is not completely correct.

As requested, we have performed statistical testing on the box plots in question (two-tailed paired t-test), given that each gene (n=88) has two expression values corresponding to either control KD or Yap1 KD. Accordingly, we obtained a P-value of 0.0096, indicating that the difference in expression after differentiation in control KD vs. Yap1 KD is significant (modified Figure 3—figure supplement 1D). We also performed the same test on pro-apoptotic genes as a control and obtained a P-value of 0.9141, indicating lack of significance. Both have been marked in the panels as ** (since it falls between the intervals of 0.01 and 0.001) and NS, respectively.

5) In the same section; "Intriguingly, analyzing publicly available RNA-seq data from four human cell types using gene set expression analysis (GSEA) showed dysregulation of genes involved in apoptotic signaling after YAP1 KD relative to control KD, showing that Yap1 may transcriptionally regulate apoptosis in both mouse and human cells (Figure 2—figure supplement 2D). Is the GSEA overlaps significant in Figure 2—figure supplement 2D? What are the p and q values? If it is not significant, then the statement is not completely correct.

We have checked the p- and q-values. Unfortunately, we found that only the HCC364 (lung cancer) cell line showed significant dysregulation. Its p- and q-values were 0.029 and 0.0325, respectively, whereas the values were generally >0.49 or higher in the other cancer cell types. We thus contend that our initial claim is not justified by the data without performing a much more comprehensive analysis or cancer-related experiments, which would be outside the scope of this manuscript, and thus we propose removal of that panel (Figure 3—figure supplement 1F) in our revised manuscript.

6) Subsection “Yap1 directly regulates apoptosis-related genes via transcription”: "Meanwhile, transient OE of Yap1 led to higher luciferase activity with anti-apoptotic gene regulatory elements and lower luciferase activity with pro-apoptotic gene regulatory elements (Figure 3E)." What happens to endogenous Bcl2 and BCl^-^xl genes in YAP1 overexpressing cells? They should be prominently upregulated, if the authors model is correct?

Thank you for this comment. That is indeed what is expected. As was mentioned in response to comment 1-2, we performed transient OE of Yap1 and found a modest increase in expression of anti-apoptotic factors (new Figure 3—figure supplement 1C).

One may question why the effects of Yap1 OE were modest and generally did not exceed two-fold changes in expression. We surmise that this is because Yap1 is already relatively highly expressed in WT dESCs, and since Yap1 is merely a co-regulator that can only function in complex with several other factors that then mediate DNA binding and histone modification, we propose that limiting factors of this complex may restrict the ability for excess Yap1 to strongly upregulate its targets. In addition, apoptosis-related genes must be tightly regulated by the cell by a myriad of mechanisms to prevent tumorigenesis; just as an example, a miRNA known as miR-136-5p represses BCl^-^2 (Li et al., 2017) and miR-302b represses MCl^-^1 (Khodayari et al., 2016).

7) Figure 3D. Is TEAD-YAP1 -binding necessary for upregulation of Luciferase expression in Figure 3D? This can be analyzed by mutating TEAD-YAP-binding consensus sequences in these constructs. If YAP1 role is direct, TEAD-YAP binding to the promoter should be critical.

This is an excellent point. Indeed, a careful search of the apoptosis-related enhancers showed that almost all of them had at least one occurrence of the Tead binding consensus sequence, GG(A/T/C)AT, or its reverse strand counterpart. Initially, we attempted to delete this motif from several of our putative regulatory elements, but it was more difficult than expected, as most of the regulatory elements had multiple instances of this motif and their overall sequences were extremely repetitive, making site-directed mutagenesis more prone to error due to slippage.

Ultimately, we used two approaches. We performed site-directed mutagenesis on Yap1 itself, mutating Ser79 to Ala, which virtually abolishes the Yap1-Tead interaction (Schlegelmilch et al., 2011). We then performed transient OE of mutant and regular Yap1 in Yap1 KO cells in an attempt to rescue luciferase expression of our strongest two enhancers, MCl^-^1 distal and the newly created BCl^-^2 intronic tandem (mentioned in the earlier comment, 1-2). OE of unmutated Yap1 rescued luciferase expression in Yap1 KO cells to levels comparable to WT ESCs with Yap1 OE, whereas OE of Yap1 Ser79Ala was defective in rescue of either construct (new Figure 4F).

For our second approach, we decided to alter the sequence of our strongest enhancer, MCl^-^1 distal, and delete the Tead binding sequence (TBS) entirely, as there was only one occurrence of it (BCl^-^2 tandem had three). We deleted ATTCC (indicating the presence of GGAAT on the reverse strand) from the middle of the enhancer and found that it not only abrogated ability to respond to Yap1 OE in WT ESCs, but also broke enhancer function entirely (new Figure 4G). Although the positive control (MCl^-^1 distal, unmutated) showed an increase in luciferase expression that was just barely not statistically significant by our threshold (p = 0.0613), we argue that these experiments still collectively demonstrate that the Tead-Yap1 interaction is essential for the regulation of these *cis-*regulatory elements.

8) Figure 2 is not well organized. What happens to pro-apoptotic transcripts during differentiation in WT ES cells? Are they up or downregulated in comparison to non-differentiated cells? It appears that the direction of the changes in gene expression are similar in WT and YAP KO cells, but only the magnitude is different. This is not obvious the way the data are presented.

We agree with this feedback and have checked the expression of all apoptosis-related genes in WT ESCs in all 4 differentiation conditions (new Figure 3B). This reveals that overall, BCl^-^2 expression is upregulated, BCl^-^xL and MCl^-^1 expression is maintained, and pro-apoptotic factors are strongly upregulated, though the magnitude of upregulation depends on differentiation method. Thus, the direction is indeed mostly the same. This also gives more context to what appear to be only mild differences in pro-apoptotic gene expression between WT and Yap1 KO; though Yap1 KO cells only have ~2x more Puma than WT in -LIF conditions (modified Figure 3C), Puma is already being upregulated 4x during differentiation in WT ESCs to begin with (new Figure 3B).

9) Figure 3C. Is there any change in YAP1 binding to pro- and anti-apoptotic genes during differentiation? This static data are not very informative. Especially considering that some YAP1 binding genes are upregulated while others are downregulated.

Thank you for the comment. We agree that static data are generally not as informative. Before submission, we had attempted to perform ChIP-seq of Yap1 in +LIF conditions. However, we did not submit this to NCBI or present it in our manuscript, as relatively a small number of peaks were enriched over background. This was completely expected, as Yap1 is almost entirely cytoplasmic during self-renewal (Chung et al., 2016) and therefore should not be binding DNA. Due to your feedback, we have submitted this sequencing data to NCBI under the same GSE number and added the L+ ChIP-seq data to modified Figure 4B, which has also been slimmed down for clarity (removed the low Yap1 peaks, as showing the L+ data is a better control and matches the p300 data presentation better). We also reanalyzed the data overall and have updated the number of peaks in the main text.

As shown in Author response image 2, binding of Yap1 and p300 at loci MCl^-^1 and BCl^-^2 increases upon differentiation according to ChIP-seq (Author response image 2). Pro-apoptotic factors Bmf and Puma (*Bbc3*) also gain both Yap1 and p300 occupancy although not very strong. This increase in p300 occupancy, even on pro-apoptotic genes, is consistent with our RT-qPCR data during WT ESC differentiation (new Figure 3B). Yap1’s co-repressor function has been characterized previously (Kim et al., 2015). Perhaps it recruits HDAC to fine-tune the expression of pro-apoptotic genes during differentiation (since p300 is also present), but exploring this further to determine how Yap1 knows whether to co-repress or co-activate targets on binding may be outside the scope of this manuscript.

To validate this ChIP-seq data, we decided to perform ChIP-qPCR for our two strongest enhancers (corresponding to the two genes that seem to have the strongest phenotypic effect), matching our additional luciferase assays in new Figures 4F and 4G. Indeed, ChIP-qPCR confirms a significant enrichment of Yap1 at BCl^-^2 and MCl^-^1’s regulatory elements upon differentiation (Author response image 2). Collectively, these data plus our luciferase assays, RT-qPCR, and other experiments demonstrate Yap1’s context-dependent role in maintenance of MCl^-^1 expression and upregulation of BCl^-^2.

**Author response image 2. respfig2:** (**A**) Signal tracks indicating Yap1 (red) and p300 (blue) ChIP-seq performed in ESCs (+LIF for Yap1, 2i for p300) and dESCs (-LIF for Yap1, EpiLC for p300). Grey bars indicate regions during primer design for ChIP-qPCR. (**B**) ChIP-qPCR of regulatory elements associated with apoptosis-related genes in +LIF (black) and -LIF (grey) FLAG-Bio-Yap1 ESCs (n = 2). Stars indicate p-values of 0.0015 and 0.0066, respectively (two-tailed t-test).

10) Figure 4C should have data for Bmf and Puma knockdowns in WT cells. If the difference between WT and YAP1 KO cells significantly reduced upon Bmf and Puma knockdowns in both cell types, then Bmf and Puma genes play a specific role. If the difference between WT and KO cells is not significantly reduced upon Bmf and Puma knockdowns in both cell types, then Bmf and Puma genes do not play a specific role.

We thank the reviewer for this comment. Pro-apoptotic factors are somewhat redundant in their function. KD of Bmf or Puma reduced cell death in Yap1 KO cells because the abnormally high activation of intrinsic apoptosis in such cells is easily perturbed by reducing the pool of available Casp9 (old Figure 1I) which is downstream of such factors. Due to this comment, we performed KD of the same factors in WT, and we found that KD did not consistently reduce cell death in WT between constructs (modified Figure 5D). This is consistent with the rest of our data that rates of apoptosis in WT are typically quite robust and difficult to decrease. Thus, Bmf and Puma do play a specific role in Yap1 KO cells.

11) Figure 1—figure supplement 1J. Knockdown of what gene?

This panel was referring to knockdown of Casp9. To increase clarity, this has been indicated to the panel below the x-axis. Due to figure reorganization, this is now modified Figure 1—figure supplement 1H.

Reviewer #2:

[…] Overall this is an interesting study, however some important issues need to be examined and clarified before publication:

We thank the reviewer for critically evaluating our manuscript. We hope that our additional experiments and clarifications offer sufficient evidence to justify our claims.

1) Please show rescue of phenotype in YAP1 KO ESCs upon ectopic expression of Yap1 in order to exclude off target phenotypes.

We have already performed the rescue experiment of ectopic expression of Yap1 in Yap1 KO ESCs, as this was the positive control for our BCl^-^xL OE experiment in old Figure 6A. We had generated three stable Yap1 KO cell lines overexpressing Yap1 (old Figure 6—figure supplement 1A) and performed the cell death assay (old Figure 6A) showing reduction of cell death during differentiation to levels below WT ESCs just like BCl^-^xL OE.

2) Does overexpression of YAP1 in WT ESCs further decreases apoptosis upon LIF removal?

We thank the reviewer for this comment, as we did not sufficiently address whether excess Yap1 can augment survival. We generated three stable WT ES cell lines overexpressing Yap1 (new Figure 1—figure supplement 1F) and performed the cell death assay, showing reduction of cell death during differentiation to levels below WT ESCs (~10%) (new Figure 1C).

3) What about the other hippo pathway effector – can TAZ overexpression rescue the phenotype of YAP1 KO? Does TAZ overexpression in WT ESCS reduce apoptosis upon LIF withdrawal?

We greatly appreciate this comment, because the literature has shown both overlapping and distinct functions for Yap1 and its paralog, Taz (Hong et al., 2005; Matsushita et al., 2018; Plouffe et al., 2018). We inducibly overexpressed Taz and this not only rescued the phenotype of Yap1 KO cells, but also slightly reduced cell death in WT ESCs from ~30% to 25% (new Figure 6C and Figure 6—figure supplement 1E). Though this reduction in cell death was consistent enough compared to uninduced cells to be statistically significant, it is not a dramatic reduction, and OE of Yap1 in WT ESCs was more effective in reducing cell death (new Figure 1C). Taz OE increases BCl^-^xL protein levels but does not seem to affect MCl^-^1 (new Figure 6D). Thus, excess Taz phenotypically rescues complete lack of Yap1 during differentiation.

4) The authors indicate "Thus, Yap1 is key for survival during ESC differentiation, and ablation of Yap1 exacerbates apoptosis specifically rather than triggering an alternate cell death pathway." However, they do not show survival rate of yap KO in a few passages in differentiated state. How do they know lethality is during the differentiated state transition, and not just more lethal in the differentiated state itself?

This is an intriguing comment, as we did not consider whether Yap1 had any roles during later differentiation, as most Yap1 studies occur in cancer cells or at very specific developmental stages. We decided to split the question into two parts. First, do Yap1 KO cells survive if they are continuously cultured? Second, does depletion of Yap1 after the exit from self-renewal influence apoptosis?

We found that by d7 of differentiation, cell death rates in Yap1 KO remained high (new Figure 2F). Some cells remained in somewhat round, small colonies, consistent with the differentiation defect phenotype previously reported (Chung et al., 2016), whereas other cells attained a flattened, stretched morphology similar to WT dESCs at this stage (data not shown). We did not passage the cells further, as even WT dESCs became sensitive to dissociation and replating at this stage, which would have obfuscated any subsequent cell death assays.

Secondly, we used a well-characterized inhibitor of Yap1, verteporfin, at various stages of differentiation (new Figure 2E) and found that inhibition was most deleterious during the exit from self-renewal (new Figure 2F). By d7, even the highest dosage had no effect on cell survival, and WT dESCs were relatively healthy at this point. Collectively, we propose that with regards to apoptosis, Yap1’s importance is primarily during the transition period from self-renewal to differentiation, and a few Yap1 KO dESCs do manage to survive and continue growing.

5) The authors indicate "This implied that the abnormally high rates of apoptosis in Yap1 KO cells are sustained by heightened Casp9 activation. However, since mRNA expression of caspases was relatively equal between Yap1 KO cells and WT cells during differentiation, we speculated that Yap1 may regulate other factors that indirectly affect the rate of caspase cleavage". This is over speculation. If mRNA levels are the same, protein levels should be tested.

To address this comment, we have performed Western blots for uncleaved Casp3 as well as uncleaved Casp8 and its first processed fragment (p43), and we had already performed Casp9 Western blot (modified Figure 1G). For Casp3, levels fluctuate wildly during differentiation, but the pattern remains roughly the same. However, for both Casp8 and Casp9, total protein levels are higher in Yap1 KO cells. As this does not reflect mRNA expression, we speculate that there must be some post-translational mechanism involved, and we do not deny that this may contribute to the enhanced cell death observed in Yap1 KO dESCs.

However, we believe that cleavage of caspases is generally more critical for caspase activity than relative protein levels. For example, Casp9 cleavage relieves inhibition of its activity by *XIAP*, allowing it to be fully active in the apoptosome holoenzyme complex (Twiddy and Cain, 2007). Thus, we believe that our focus on upstream regulators of mitochondrial outer membrane integrity (which affect apoptosome formation and thus Casp9 activation) is reasonable.

6) The authors indicate "Immunocytochemistry shows that Yap1 nuclear localization tends to be associated with higher expression of BCl^-^2 in -LIF, but not in +LIF conditions, where undifferentiated ESCs show cytoplasmic Yap1 and weaker BCl^-^2 staining" (Figure 2—figure supplement 2A)". The ICC looks of poor quality and looks like a multicellular effect and not a single cell phenotype. Can this be explained or improved?

We agree that the ICC previously presented did not provide sufficient evidence for our claims. We redesigned the experiment to make it more rigorous; instead of looking at +LIF and -LIF conditions, we compared WT and Yap1 KO ESCs, both in -LIF. Our new ICC shows reduced BCl^-^2 and MCl^-^1 in Yap1 KO dESCs compared to WT as quantified by ImageJ (new Figure 3—figure supplement 1A and B). As a form of antibody validation, we also stained for the mitochondria using a MitoTracker dye, verifying that our anti-apoptotic proteins colocalized with the mitochondria, especially MCl^-^1 (weighted colocalization coefficient of BCl^-^2 or MCl^-^1 with the mitochondria, ~0.7-0.9). Additionally, we found reduced mitochondrial content in Yap1 KO dESCs. This is consistent with prior observations that ESCs increase mitochondrial biogenesis during differentiation (Wanet et al., 2015); thus, it is logical that defects in differentiation may hamper this process.

7) The authors indicate "Collectively, these data show that Yap1 is critical for proper BCl^-^2 induction during differentiation". However Yet in Figure 3E they show no change in BCL2 luci with OE of YAP? Can this be better explained and addressed.

We are thankful for this comment. Reviewer 1 brought up a similar point (comment 1-2). We believe that testing Yap1-bound putative regulatory elements does not always yield a functional enhancer that behaves as it should in cells. To address this contradiction, we created a tandem enhancer composed of our old BCl^-^2 intronic regulatory element and another Yap1-occupied sequence (which also possessed a Tead binding site), and the schematic of this is seen in new Figure 4—figure supplement 1H. This tandem enhancer had nearly identical basal activity to the old enhancer, but upon Yap1 OE, its activity increased 2x, consistent with our RT-qPCR data (modified Figure 4E).

8) The authors indicated "Using RNA-seq data from a previous study (Chung et al., 2016), we found that differentiation induces the expression of a group of anti-apoptotic genes in WT cells, but this induction is debilitated after Yap1 KD (Figure 2—figure supplement 2B)." No t-test performed for any of these boxplots in Figure 2—figure supplement 2. This should be added and verified.

We have performed t-tests for the boxplots in previous Figure 2—figure supplement 2 (now modified Figure 3—figure supplement 1D and E), and only found one comparison (Yap1 KD vs. control KD during differentiation) was significant. Thus, the data support our initial assertion.

9) The authors indicate "Additionally, we confirmed a physical interaction between Yap1 and Tead4 during ESC differentiation using co-immunoprecipitation (Figure 3—figure supplement 3D)." Did the authors conduct an IP for p300 as well, if they show a gene correlation in ChIP? This is also what they present in the end Figure (5), but they don't show any IP for it to substantiate this claim?

We thank the reviewer for this feedback. We have performed a Co-IP and confirmed a physical interaction between Yap1 and p300 (modified Figure 4—figure supplement 1B) in dESCs, which supports our contention of Yap1 and p300 co-occupancy according to our ChIP-seq data.

10) The authors indicate "Furthermore, deletion of Yap1 significantly sensitized ESCs, but not undifferentiated ESCs, to loss of Δѱ in response to BH3 mimetic treatment. We then investigated whether the higher loss of Δѱ in Yap1 KO cells correlated with greater rates of cell death". I wonder why did the authors use only response to BH3 treatment? Why not +/- LIF like before? Could this be also shown and explained.

We appreciate this comment; initially, we only used BH3 treatment because most assays measuring Δѱ in the literature used either BH3 mimetics or fragments of BH3-only proteins. We have performed the mitochondrial membrane potential assay in both WT ESCs and Yap1 KO ESCs during 4 different differentiation and 2 different self-renewal conditions (Figure 5A). Loss of Δѱ as measured by A525/A590 was 1.5 to 2-fold higher in Yap1 KO ESCs compared to WT ESCs in all differentiation conditions tested, but not in ES+ or 2i media. This supports our BH3 mimetic data, and collectively, the results show that mitochondria in Yap1 KO cells are more sensitive to depolarization.

Reviewer #3:

[…] This study is well executed and timely, and the message is clear and significant in contributing to our understanding of the functions of Yap1 during ESC differentiation and early development and of the balance control between survival and death during cell fate changes. While I am strongly supportive for its publication in eLife, I do have some major and minor points listed below that could further enhance the clarity and improve the manuscript. And the authors should pay more attention to the labeling in the figures and provide a few minimal additional experiments and more explanations to the experiment design to make the overall story more straightforward and clearer.

We are grateful for the reviewer’s comments and overall support of our manuscript. Following this feedback and that of other reviewers, we have performed additional experiments, reorganized several of our figures, and fixed the labeling to improve the flow and clarity of the story.

Major points:

*1) At beginning the authors tested several mESC differentiation models (LIF withdraw, N2B27, and IDE1). However, it will be nicer if they also test cell apoptosis of WT/Yap1KO ESCs in EpiLC differentiation, because 1) EpiLC differentiation mimics the* in vivo *epiblast maturation at pre-to-post implantation stage, and 2) it is more related to the context of Yap1 and p300 ChIP-seq analysis as performed in Figure 3.*

Thank you for this comment. We have performed not only the cell death assay during EpiLC differentiation (verification of markers shown in new Figure 2—figure supplement 1C), but we also repeated a couple of other experiments related to the various differentiation methods (RT-qPCR and JC-10 assay).

In EpiLC conditions, Yap1 KO dESCs experience significantly higher cell death, and this can be rescued almost completely by zVAD (new Figure 2D). They also have a similar defect in anti-apoptotic gene expression (modified Figure 3C) and increased mitochondrial depolarization during differentiation (new Figure 5A). Thus, even in EpiLC differentiation (which mimics epiblast formation), Yap1 is instrumental for cell survival.

2) Figure 1 shows the upregulation of Casp and apoptosis in Yap1 KO starts early (24-48 hrs) after LIF withdrawal, but Figure 2A and 2B show the upregulation of anti-apoptotic factors (BCl^-^2, BCl^-^xL and MCl^-^1) in WT ES starts late (72 hrs) after LIF withdrawal. BCl^-^2 is the regulator upstream of Casp. The authors should provide some explanations about these results.

We appreciate this feedback, as the issue of timing of expression differences vs. the activation of the apoptotic cascade does require more explanation. Although BCl^-^2 is critical for preventing caspase activation, we would like to emphasize that MCl^-^1 is also important. Even though Yap1 KO cells have only a mild (~50%) reduction in MCl^-^1 compared to WT ESCs, it is much more highly expressed than BCl^-^xL or BCl^-^2. Loss of MCl^-^1 leads to the death of undifferentiated ESCs (Huskey et al., 2015). Although the same authors contend that BCl^-^2 and BCl^-^xL gain in importance during later stages of differentiation, we reason that because apoptosis starts so early (24-48 hr after -LIF), MCl^-^1 is still essential for survival in early differentiation. This is supported by our BH3 mimetic data showing that ESCs and dESCs (at 48 hr) are exquisitely sensitive to MCl^-^1 inhibition (old data in current Figure 5C) and KD on d1 potentiates cell death in dESCs by d3 (old data in current Figure 6E).

To address whether MCl^-^1 plays a role in inhibiting caspase activation during early differentiation, after 1 day of -LIF, we acutely inhibited either MCl^-^1 using a selective inhibitor or BCl^-^2/BCl^-^xL/BCl^-^w using a different inhibitor in both WT and Yap1 KO cells. Treatment lasted for only 4 hours until harvest of protein lysates.

The results in new Figure 5F show that inhibition of MCl^-^1 strongly increases Casp3 cleavage in both WT and Yap1 KO ESCs. We confirmed Yap1 KO cells have less MCl^-^1 protein than WT cells do, even just 28 hr after -LIF, and they are much more sensitive to inhibition of BCl^-^2/BCl^-^xL/BCl^-^w than WT ESCs. This is logical, since all of these proteins collaborate at the mitochondrial outer membrane to prevent apoptosis, and they are known to compensate for one another in chemotherapy-related studies (Moujalled et al., 2018).

Thus, we contend that lower expression of the most abundant anti-apoptotic protein, MCl^-^1, in Yap1 KO cells contributes to an increase in caspase cleavage that is already apparent early in differentiation, even before the defect of BCl^-^2 expression is particularly noticeable.

3) In the Figure 1E, 1F and 2D, please indicate the time after LIF withdrawal (or the time of induced differentiation) in hours to better compare them with other results (like Figure 1D, 2A, 2B). I didn't find their time point information in Figure/Figure legends/Text/Experimental procedures.

We have added the appropriate time of differentiation for the aforementioned panels directly to the figures.

4) About Figure 3D and 3E: (1) the authors should at least provide one representation of the reporter construct;

As requested, we have provided a representation of the firefly reporter construct in new Figure 4—figure supplement 1G.

(2) the authors should indicate the regulatory elements (e.g., by using bars) used in the luciferase constructs in the Yap1 and p300 ChIP-seq tracks;

The grey bars currently on the ChIP-seq tracks do indicate the regulatory elements used in the luciferase constructs. However, since this is not indicated in the panel and the bars are extremely light, it is indeed confusing, and the width of the bars is much larger than the sequence actually used for cloning. Thus, we have slimmed down the grey bars, made them darker, and added a key to reduce any misinterpretation (modified Figure 4C).

(3) I am curious to see the expression level of Yap1 in Yap1 OE: was it rescued to a WT level or an overexpression?

We apologize for any confusion. We used WT ESCs in modified Figure 4F, so this would be an overexpression. Though we did not perform RT-qPCR to determine the fold change in Yap1 expression after 24 hr of transfection when the assay was performed, you may refer to new Figure 3—figure supplement 1C, where Yap1 OE was measured after 48 hr of transfection. There was a 16x increase in mRNA levels, which establishes an upper bound. We expect that the actual increase in mRNA levels during the luciferase assay would be quite a bit lower, as only 24 hr had elapsed after transfection, and the amount of DNA transfected was 1/3 of the normal amount, as 3 vectors were transfected simultaneously.

*5) Figure 3*– *data are clear but too shallow/superficial: According to the increased p300 (and H3K27ac) ChIP-seq intensities at Yap1 peaks during differentiation, it is implied that Yap1 may function as an activator of transcription for the anti-apoptotic genes BCl^-^2, BCl^-^xL, MCl^-^1. ChIP-qPCR are required to confirm the Yap1 binding at those loci (Figure 3C). Furthermore, authors should also compare the Yap1 and p300 ChIP intensity between ES and dES situations, to support their conclusion that Yap1 activates the anti-apoptotic genes during differentiation.*

This is an excellent point that reviewer 1 also brought up. Please refer to Author response image 2 in comment 1-9. In sum, we in turn performed ChIP-seq and the data showed that both Yap1 and p300 occupancy increase on apoptosis-related genes during differentiation. ChIP-qPCR showed enrichment for BCl^-^2 and MCl^-^1. Collectively, our luciferase, RT-qPCR, immunoblot, ICC, and now ChIP-qPCR data all agree that Yap1 activates BCl^-^2 and MCl^-^1 upon differentiation, but we have less evidence for its direct regulation of the other apoptosis-related genes.

Does Yap1 physically interact with p300? If so, the authors may want to show if Yap1 recruits p300 to chromatin.

Immunoblot shows that pulldown of FLAGBio-Yap1 also brings down p300, showing that a physical interaction between the two factors does indeed exist during ESC differentiation (Figure 4—figure supplement 1B). Although we intended to do ChIP-qPCR of Yap1 KO cells during differentiation to test whether ablation of Yap1 reduced p300 recruitment to Yap1 peaks, it was difficult to obtain a sufficient quantity of cells during differentiation for ChIP, which requires a lot of starting material for the fixation step. We considered adding zVAD to boost survival and cell yield so that we could do ChIP on Yap1 KO dESCs, but we were concerned that this may have some unintended effects. Therefore, we did not test whether Yap1 actively recruits p300 to chromatin. However, it is known from previous literature investigating YAP1 occupancy in human SF268 glioblastoma cells that KD of Yap1 reduced p300 occupancy as well as H3K27ac at Yap1’s peaks (Stein et al., 2015). Our Co-IP is thus consistent with prior observations and it is likely that Yap1 recruits p300, although we did not explicitly test that hypothesis.

6) Figure 4C: the label is misleading. In the figure, the authors should indicate the Bmf KD and Puma KD are performed in Yap1 KO.

Thank you for pointing this out. We have additionally performed Bmf and Puma KD in WT ESCs (modified Figure 5D) in response to a comment brought up by reviewer 1 using the same shRNA constructs. We also have added the label to modified Figure 5—figure supplement 1B as suggested.